# A toolbox of astrocyte-specific, serotype-independent adeno-associated viral vectors using microRNA targeting sequences

Amy J. Gleichman [1], Riki Kawaguchi[2], Michael V. Sofroniew [3] & S. Thomas Carmichael [1] ✉

Astrocytes, one of the most prevalent cell types in the central nervous system (CNS), are critically involved in neural function. Genetically manipulating astrocytes is an essential tool in understanding and affecting their roles. Adeno-associated viruses (AAVs) enable rapid genetic manipulation; however, astrocyte specificity of AAVs can be limited, with high off-target expression in neurons and sparsely in endothelial cells. Here, we report the development of a cassette of four copies of six miRNA targeting sequences (4x6T) which triggers transgene degradation specifically in neurons and endothelial cells. In combination with the GfaABC1D promoter, 4x6T increases astrocytic specificity of Cre with a viral reporter from <50% to >99% in multiple serotypes in mice, and confers astrocyte specificity in multiple recombinases and reporters. We also present empty vectors to add 4x6T to other cargo, independently and in Cre/Dre-dependent forms. This toolbox of AAVs allows rapid manipulation of astrocytes throughout the CNS, is compatible with different AAV serotypes, and demonstrates the efficacy of using multiplexed miRNA targeting sequences to decrease expression in multiple off-target cell populations simultaneously.

Astrocytes comprise roughly 20–40% of the cells in the CNS and are found across all brain regions[1]. They play crucial roles in both physiological and pathological states and across the lifespan[2]. The ability to specifically genetically manipulate astrocytes is critical in order to understand and alter astrocytic functions; to this end, viral vectors can offer experimental flexibility, speed, and selectivity[3]. Adeno-associated viruses (AAVs) are widely used viral vectors due to their safety profile, limited toxicity, and the natural occurrence of serotypes with different tropism. Furthermore, engineered AAV capsids can confer new properties on the virus. For example, newly developed AAV capsids such as PHP.eB[4] are systemically deliverable; these viruses can be injected into mice intravenously (IV), cross the blood–brain barrier, and transduce cells throughout the CNS with a single injection.

Implementing these new vectors in astrocytes, however, is complicated by non-astrocytic expression, despite the use of astrocyte-specific promoters. This expression is variable and depends on the capsid used[5] and the cargo expressed[6,7], but extensive neuronal expression can be observed and complicates the interpretation of results[6,8–10]. This is particularly problematic when delivering Cre recombinase, which requires only low expression levels for effective genetic recombination. One way to obtain serotype-independent specificity in viral vectors—that is, specificity that is compatible with different capsids—is through the use of microRNA (miRNA) targeting sequences[11]. These targeting sequences are DNA sequences that will bind to specific miRNAs that are highly expressed in off-target cells but not in the cells of interest: in off-target cells, transgene mRNA will bind miRNAs and be degraded, but will be preserved in target cells and

[1]Department of Neurology, David Geffen School of Medicine at University of California—Los Angeles, Los Angeles, CA, USA. [2]Center for Neurobehavioral Genetics, Semel Institute for Neuroscience and Human Behavior, University of California Los Angeles, Los Angeles, CA, USA. [3]Department of Neurobiology, University of California—Los Angeles, Los Angeles, CA, USA. ✉e-mail: SCarmichael@mednet.ucla.edu

allow expression. One benefit to this approach to cell-type specificity is that these sequences can be used within the recombinant AAV genome, rather than the capsid, and therefore can improve specificity across serotypes. Using miR-124, this approach increased astrocyte selectivity in lentiviruses[12] and in AAVs for expression of GFP[8,13] and Cre[8]. While miR-124 targeting improves astrocyte specificity, off-target expression in neurons and endothelial cells was still readily observed, particularly with systemic Cre delivery[8]. This suggests that while miR targeting holds promise for astrocytic viral specificity, there is room for improvement, particularly when delivering recombinases.

Here, we report the development of a cassette of miRNA targeting sequences, comprised of four copies of each of six miRNAs, to de-target neurons and endothelial cells simultaneously and improve astrocytic specificity in AAV vectors. This cassette increases astrocyte specificity from <50% to >99% and is effective across multiple serotypes and a variety of cargo, including multiple recombinases. The efficacy of this cassette does not decrease over time and induces minimal transcriptomic changes in transduced astrocytes. By including this cassette on a range of plasmids, we present a toolbox of vectors, including recombinases, reporters, and empty vectors, for rapid implementation of this approach, allowing for improved options for astrocyte-specific viral manipulation.

## Results

### Development of miR targeting cassette to improve astrocyte specificity

To assess the astrocytic specificity of Cre viral delivery using current vectors for systemic delivery, we administered PHP.eB viruses carrying Cre under a glial fibrillary acidic protein (GFAP) promoter (PHP.eB::GFAP-Cre; Addgene 105550); GFAP is an astrocyte-specific protein and elements of the promoter have been widely used to target astrocytes[14]. We also generated a vector using the smaller GfaABC1D promoter[15] (PHP.eB::GfaABC1D-Cre). As only low levels of Cre are necessary for recombination, we omitted a woodchuck hepatitis virus post-transcriptional regulatory element (WPRE), used to increase transgene expression, reasoning that this might decrease off-target transgene expression; a similar approach was recently used to improve targeting of Muller glia[16]. Viruses were delivered systemically using retroorbital administration into young adult (2–5 months) C57BL/6J mice. To assess Cre activity, we co-injected a Cre-dependent GFP reporter (PHP.eB::CAG-flex-GFP, Addgene 28304); 2 weeks post injection, we evaluated colocalization of GFP+ cells, enhanced with an anti-GFP antibody, with the astrocytic nuclear marker Sox9[17] in the cortex. Both promoters were predominantly non-astrocytic, although GfaABC1D showed higher astrocyte selectivity (Fig. 1a; Sox9+ GFAP: 22.79% ± 1.50 vs GfaABC1D: 38.81% ± 3.64); the majority of Sox9-GFP+ cells were neurons (NeuN+). Viral Cre-dependent flex constructs can show Cre-independent expression[18]; therefore, we repeated PHP.eB::GfaABC1D-Cre delivery in Ai14-tdTomato mice[19], a transgenic line with Cre-dependent tdTomato expression. While astrocyte specificity was greater in these mice, we still observed extensive neuronal (NeuN+tdTomato+, 29.16% ± 2.78) and minimal endothelial (CD31+ tdTomato+, 1.23% ± 0.07) contamination (Fig. 1b).

To decrease neuronal expression, we added a previously reported cassette with four copies of miR-124-3p targeting sequence (miR124T.4x[12]) immediately following the stop codon terminating GfaABC1D-Cre (Fig. 1c). We also identified four other miRNAs−miR-137-3p, miR-329-3p, miR-369-5p, and miR-431-3p−with similar patterns of high neuronal/low astrocytic expression[20,21] and generated viral vectors for each that included two copies of the targeting sequence. We delivered these vectors systemically using PHP.eB, co-injecting with PHP.eB::CAG-flex-GFP, and co-stained for GFP and Sox9. As expected, miR124T.4x showed high levels of astrocyte specificity (95.39% ± 1.038 Sox9+). miR137T.2x and miR431T.2x showed lower, but significant, increases in astrocyte specificity (miR137T.2x: 56.37% ± 1.85 Sox9+;

miR431T.2x: 58.72% ± 5.85 Sox9+). While miR329T.2x and miR369T.2x did not show a significantly different proportion of astrocytic specificity, they appeared to trend towards greater specificity (miR329T.2x: 47.31% ± 2.62 Sox9+; miR369T.2x: 49.15% ± 6.06 Sox9+) (Fig. 1d); notably, these increases in specificity seen with the four previously untested miRs were observed with only two copies of their relative targeting sequences, to facilitate rapid sequence synthesis. Therefore, in order to target astrocytes as specifically and robustly as possible, we developed a multiplexed vector with four interspersed copies of the targeting sequences for each miR. To de-target endothelial cells, we also included four copies of a targeting sequence for miR-126a-3p[22], generating a final cassette of four copies of each of six miRNA targeting sequences with a final length of 529 nucleotides (4x6T, Fig. 1e).

We systemically injected PHP.eB::GfaABC1D-Cre-4x6T into wild-type mice with PHP.eB::CAG-flex-GFP (5x10^11vg/mouse per virus) and evaluated astrocyte specificity in the cortex. Compared to PHP.eB::GfaABC1D-Cre-miR124T.4x, the presence of additional miRNA targeting cassettes significantly improved astrocyte specificity (Fig. 1f; miR124T.4x, 95.39% ± 1.04 Sox9+ vs 4x6T, 98.87% ± 0.06 Sox9+). To test the specificity of PHP.eB::GfaABC1D-Cre-4x6T in transgenic animals, we injected the virus into Ai14-tdTomato mice (5x10^11vg/mouse); this yielded even higher astrocyte specificity compared to PHP.eB::CAG-flex-GFP reporter (Fig. 1g; Ai14: 99.75% ± 0.04 Sox9+ vs flex-GFP, 98.87% ± 0.06 Sox9+). This reflected successful de-targeting of both neurons and endothelial cells (Fig. 1g; 284.7-fold reduction in % tdTomato+NeuN+/tdTomato+ cells; 13.9-fold reduction in % tdTomato+CD31+/tdTomato+ cells). Further, the relatively lower astrocyte specificity seen with PHP.eB::CAG-flex-GFP compared to Ai14-tdTomato could be improved by adding the 4x6T cassette to a viral flex reporter (membrane-targeted V5 spaghetti monster reporter[23], PHP.eB::CAG-flex-lck-smV5-4x6T; "flexV5-4x6T") (Fig. 1g; 99.81% ± 0.04 Sox9 + cells), thereby adding miR targeting specificity to both the Cre and reporter components of the expression system.

### Role of titer and serotype in maximizing astrocyte specificity

While astrocyte labeling with PHP.eB::GfaABC1D-Cre-4x6T was highly specific, it did not transduce all astrocytes. To improve the percentage of astrocytes labeled, we injected PHP.eB::GfaABC1D-Cre-4x6T systemically at a higher titer ($3 \times 10^{12}$ vg/mouse). This titer maintained 99.67% ± 0.12 astrocyte specificity while labeling 65.66% ± 1.38 of astrocytes in the cortex (Fig. 2a). To label a greater percentage in a specific area, we injected virus intracortically (Fig. 2b). However, intracortical injection of high titer (500 nl of $1 \times 10^{12}$ vg/ml per virus) of PHP.eB::GfaABC1D-Cre-4x6T with PHP.eB::CAG-flex-lck-smV5-4x6T showed extensive neuronal transduction (Fig. 2c), despite the presence of the 4x6T cassette on both viruses. Therefore, we repackaged the reporter construct in the more astrocyte-selective AAV2/5 serotype[5] (AAV2/5::CAG-flex-lck-smV5-4x6T) and co-injected PHP.eB::GfaABC1D-Cre-4x6T and AAV2/5::CAG-flex-lck-smV5-4x6T into Ai14-tdTomato mice. High-titer intracortical injection transduced all astrocytes within the core of the injected region but with relatively lower astrocyte specificity (Fig. 2d, 87.19% ± 3.97 tdTomato+Sox9+/tdTomato+). Using the AAV2/5 reporter, however, maintained astrocyte selectively even at high titers (Fig. 2d, 99.75% ± 0.09 V5+Sox9+/V5+). Thus, it is possible to label a high percentage of geographically restricted astrocytes through direct injection of the virus into the brain by combining capsid choice and the inclusion of the miR targeting cassette on the delivered gene.

To evaluate more broadly how serotype affects astrocyte specificity with 4x6T, we packaged GfaABC1D-Cre and GfaABC1D-Cre-4x6T in AAV2/1, AAV2/5, and AAV2/9, and injected each virus intracortically with AAV2/5::CAG-flex-lck-smV5-4x6T into Ai14-tdTomato mice. In the absence of the 4x6T cassette, all Cre viruses showed high levels of tdTomato+ neuronal contamination, although AAV2/5 showed the highest astrocyte specificity (AAV2/1: 21.48% ± 3.39 Sox9+; AAV2/5: 49.17% ± 4.21 Sox9+; AAV2/9: 27.67% ± 3.49 Sox9+; Fig. 2e). Adding the

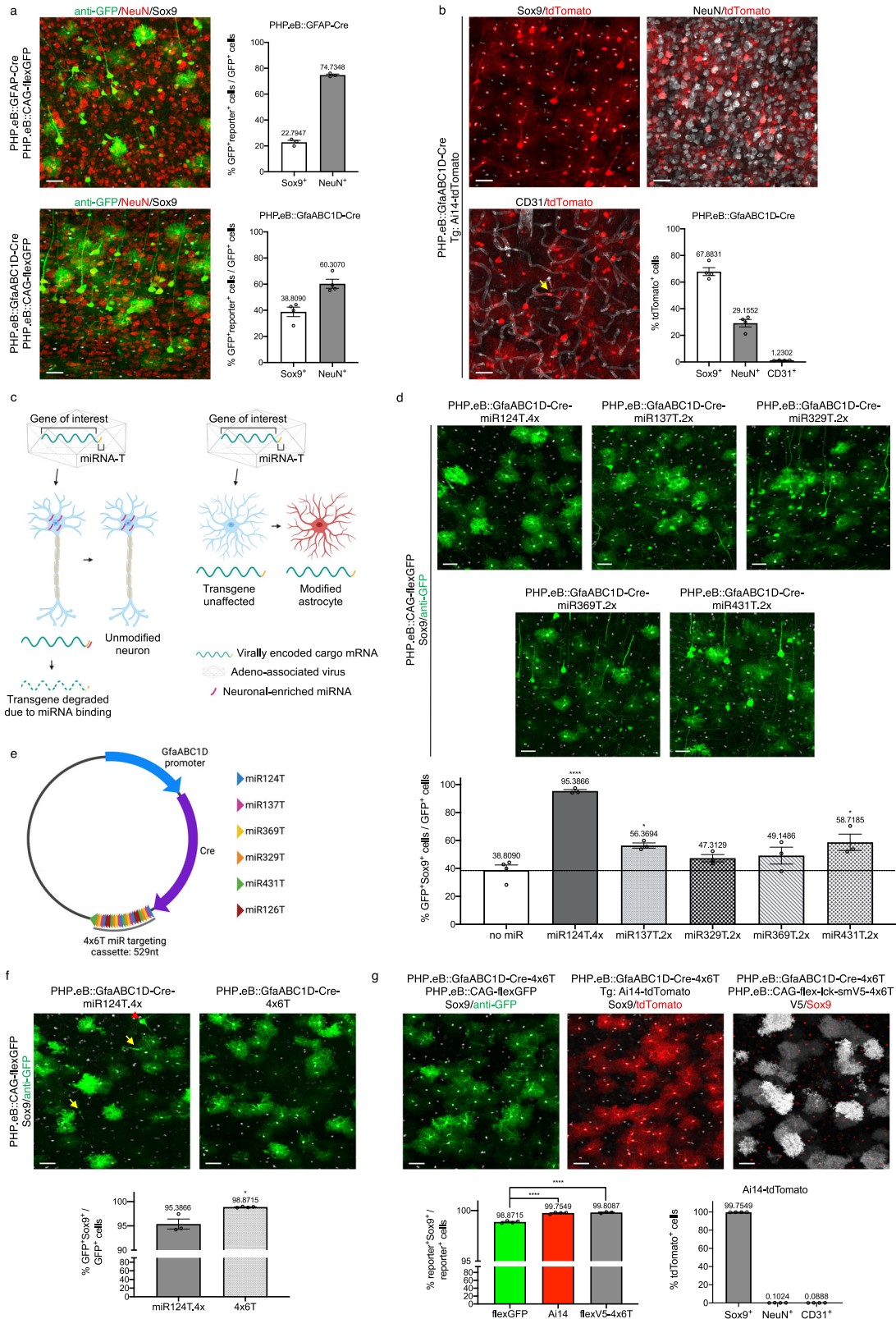

4x6T cassette to Cre increased astrocyte specificity of tdTomato expression across all serotypes (AAV2/1: 87.20% ± 1.67 Sox9+; AAV2/5: 95.51% ± 0.95 Sox9+; AAV2/9: 90.89% ± 1.11 Sox9+; Fig. 2e).

Including the 4x6T cassette on the viral reporter (AAV2/5::CAG-flex-lck-smV5-4x6T) improved astrocyte specificity when using GfaABC1D-Cre with no miR cassette (AAV2/1-Cre: 95.27% ± 1.57 Sox9+; AAV2/5-Cre: 98.77% ± 0.84 Sox9+; AAV2/9-Cre: 96.16% ± 2.13 Sox9+; Fig. 2e). This was

further improved using both Cre-4x6T and the 4x6T reporter (AAV2/1-Cre-4x6T: 99.68% ± 0.19 Sox9+; AAV2/5-Cre-4x6T: 100% ± 0 Sox9+; AAV2/9-Cre-4x6T: 99.71% ± 0.21 Sox9+; Fig. 2e). AAV2/5 showed the highest level of astrocytic specificity in all conditions, suggesting that serotype can influence specificity, although the addition of the 4x6T cassette on both components of a fully viral Cre/flex system yields extremely high astrocyte specificity at high titers across multiple serotypes.

**Fig. 1 | Low astrocytic viral specificity can be enhanced with miRNA targeting sequences. a** Systemic delivery of two different GFAP-based promoters (GFAP, $n = 3$ mice; and GfaABC1D, $n = 4$ mice), co-injected with PHP.eB::CAG-flex-GFP, yield high levels of transduction of non-astrocytic cells. Sox9, astrocytes; NeuN, neurons. **b** Characterization of PHP.eB::GfaABC1D-Cre delivery in transgenic (Tg) Ai14-tdTomato mice: while the majority of tdTomato+ cells are astrocytes, there are high numbers of transduced neurons (NeuN+) and low numbers of endothelial cells (CD31+; yellow arrow); $n = 4$ mice. **c** Schematic diagram of miRNA targeting approach. **d** Changes in astrocytic selectivity by the addition of multiple copies of single miR targeting sequences ($n = 3$ mice); compared to GfaABC1D-Cre ($n = 4$ mice) (one-way ANOVA, Dunnett's multiple comparisons test, $P < 0.0001$, $F = 24.76$, df=18. All vs no miR: miR124T.4x, four copies, ****$P < 0.0001$; miR137T.2x, two copies, *$P = 0.0263$; miR329T.2x, two copies, $P = 0.4348$; miR369T.2x, two copies, $P = 0.2680$; miR431T.2x, two copies, *$P = 0.0118$.) The dotted line denotes astrocyte specificity in GfaABC1D-Cre with no miR cassettes. **e** Schematic diagram of final AAV::GfaABC1D-Cre-4x6T plasmid. **f** Enhanced astrocyte specificity with

PHP.eB::GfaABC1D-Cre-4x6T ($n = 4$ mice) vs PHP.eB::GfaABC1D-Cre-miR124T.4x ($n = 3$ mice), using a PHP.eB::CAG-flex-GFP reporter (two-tailed unpaired $t$ test, *$P = 0.0103$, $t = 4.002$, df=5). Non-astrocytes expressing GFP after transduction with PHP.eB::GfaABC1D-Cre-miR124T.4x include both neurons (red arrowhead) and endothelial cells (yellow arrows). **g** Further enhancement of astrocyte specificity using PHP.eB::GfaABC1D-Cre-4x6T with a transgenic Ai14-tdTomato reporter ($n = 4$ mice) or a PHP.eB::CAG-flex-lck-smV5-4x6T reporter ($n = 3$ mice) rather than PHP.eB::CAG-flex-GFP ($n = 4$ mice). Using a transgenic mouse line or including the 4x6T cassette on the flex reporter increases astrocyte specificity (one-way ANOVA, Tukey's multiple comparisons test, $P = 0.0144$, $F = 7.548$, df=10; flex-GFP vs Ai14, *$P = 0.0249$; flex-GFP vs flexV5-4x6T, *$P = 0.0259$). Mice were injected retroorbitally at 2–5 months old and euthanized 2 weeks post injection. Titers: GFAP-Cre, $5 \times 10^{10}$ vg/mouse + CAG-flex-GFP, $2 \times 10^{11}$ vg/mouse.; GfaABC1D-Cre, GfaABC1D-Cre-(all miRs): $5 \times 10^{11}$ vg/mouse + CAG-flex-GFP or CAG-flex-smV5, $5 \times 10^{11}$ vg/mouse. Source data are provided as a Source Data file. All data are presented as mean ± SEM. Scale bars: 40 μm.

## 4x6T cassette confers astrocyte specificity across the lifespan and in reactive astrocytes

One advantage of using viral systems is flexibility in delivering viruses at different ages. Therefore, we tested the astrocyte specificity of the virus delivered across the lifespan of the mouse, administering the virus systemically in pups at postnatal days 1–2 via temporal vein injection and in 28-month-old mice via retroorbital injection. We found that high astrocyte specificity conferred by the 4x6T cassette is maintained in the cortex across the mouse lifespan (P1-2: 99.68% ± 0.17 Sox9+; 28 m: 99.76% ± 0.09 Sox9+; Fig. 3a). However, one concern when using a GFAP-based promoter is that GFAP is also expressed in neural progenitor cells (NPCs); therefore, we evaluated astrocyte-specific expression in the dentate gyrus of the hippocampus, a region of ongoing neurogenesis in the mouse brain. Two weeks after virus delivery, we routinely found Sox9+V5+ radial glia in the hippocampus of P1-2 mice but not in young adult (2–5-month-old) animals (Fig. 3b).

To more fully evaluate astrocytic specificity in the dentate gyrus, we returned to high-titer injections ($3 \times 10^{12}$ vg/mouse) of PHP.eB::GfaABC1D-Cre-4x6T in Ai14-tdTomato reporter mice. Because Sox9 is a marker of both astrocytes and progenitor populations, we instead used the cytosolic astrocytic marker Aldh1l1 to label astrocytes. While Aldh1l1 can also be found in neural stem cells[24], Aldh1l1 expression in radial glia is relatively lower than Sox9[25], and Aldh1l1-CreERT2 transgenic mice show minimal neural stem cell recombination in the dentate gyrus when tamoxifen is administered in adulthood[26], suggesting it is a more accurate marker of astrocytic identity in neurogenic regions. We found a high degree of astrocyte specificity within the dentate gyrus (97.22% ± 0.21 Aldh1l1+; Fig. 3c); those tdTomato+ cells that were Aldh1l1− were found predominantly in the subgranular zone and granular layer (Fig. 3c), as would be expected for radial glia and their progeny. The relatively sparse labeling of Aldh1l1− cells in this neurogenic zone in adult mice suggests that the 4x6T cassette may de-target progenitor cells to some extent.

In response to CNS injury, astrocytes upregulate GFAP[27] and can locally proliferate[28]. Injury can also induce the proliferation and maturation of NPCs; in stroke, this results in new neurons in the peri-infarct cortex[29]. Therefore, we evaluated whether stroke affects astrocyte specificity of the 4x6T cassette. Young adult mice were systemically injected with PHP.eB::GfaABC1D-Cre-4x6T and PHP.eB::CAG-flex-lck-smV5-4x6T, received distal middle cerebral artery occlusion strokes (dMCAO) 2 weeks later, and were euthanized 1 week post-stroke; dMCAO produces a cortical infarct with extensive neurogenesis at this timepoint[29]. We examined astrocyte specificity of virally labeled cells in the cortex within 400μm of the infarct border, which encompasses essentially all proliferating astrocytes post-stroke[28]. Within the peri-infarct cortex, we found that astrocyte specificity of viral expression was maintained to a high

degree (99.07% ± 0.19 Aldh1l1+; Fig. 3d); although we did find rare nonastrocytic, morphologically distinct Aldh1l1−V5+ cells near the infarct border (Fig. 3d). These results suggest that astrocytic specificity of the 4x6T cassette is largely maintained after a major CNS insult.

## Astrocyte specificity of 4x6T cassette is preserved for long time periods and across CNS regions

In the majority of experiments, we evaluated astrocyte specificity 2 weeks post-viral delivery; while expression levels are robust at 2 weeks, they continue to increase[30]. Therefore, we assessed the stability of 4x6T astrocyte-specific expression over time. We systemically delivered PHP.eB::GfaABC1D-Cre-4x6T and PHP.eB::CAG-flex-lck-smV5-4x6T to young adult mice and assessed expression patterns 6 months later. Astrocyte specificity of 4x6T viral expression was maintained in the cortex at 6 months post injection (99.87% ± 0.04 Sox9+; Fig. 4a).

Given the potential for transduced progenitor cells, we evaluated astrocyte specificity in the hippocampus. Astrocyte specificity remained high throughout the hippocampus (98.28% ± 0.36 Sox9+; Fig. 4a), although it was significantly lower in the dentate gyrus (97.37% ± 0.49 Sox9+). Astrocyte specificity was high in the olfactory bulb, thalamus, cerebellum, and spinal cord (Fig. 4b, c); indeed, we could not identify a region in which there was overt non-astrocytic viral staining. Interestingly, white matter astrocytes in both the brain and spinal cord were largely resistant to systemic PHP.eB viral transduction.

## Transcriptomic changes observed in astrocytes with viral transduction with and without 4x6T cassette

One potential caveat to viral use is whether normal cellular processes are affected by viral transduction itself. Therefore, we evaluated astrocytic transcriptomes using RiboTag[31] mice, a form of translating ribosomal affinity purification (TRAP). A requisite ribosomal subunit is tagged using Cre, allowing immunoprecipitation (IP) of ribosomes. Ribosomally loaded mRNA can then be purified and sequenced. We generated four groups of RiboTag astrocyte mice: intracortical or systemic delivery of PHP.eB::GfaABC1D-Cre-4x6T; intracortical delivery of AAV2/5::GfaABC1D-Cre, the current gold-standard approach for astrocyte-enriched viral manipulation; and transgenic RiboTag/Aldh1l1-CreERT2 mice[32], to obtain astrocyte-specific labeling without viral transduction. These groups allow us to compare how viral transduction affects astrocytes; how the 4x6T cassette modifies those responses; and how the route of delivery impacts astrocytic transcriptomic responses.

We evaluated astrocyte specificity of RiboTag expression by immunostaining for the ribosomal tag (HA+, Fig. 5a). Aldh1l1-CreERT2 transgenic mice and systemic PHP.eB::GfaABC1D-Cre-4x6T were highly astrocyte-specific (Aldh1l1-CreERT2: 99.95% ± 0.03 Aldh1l1+; systemic

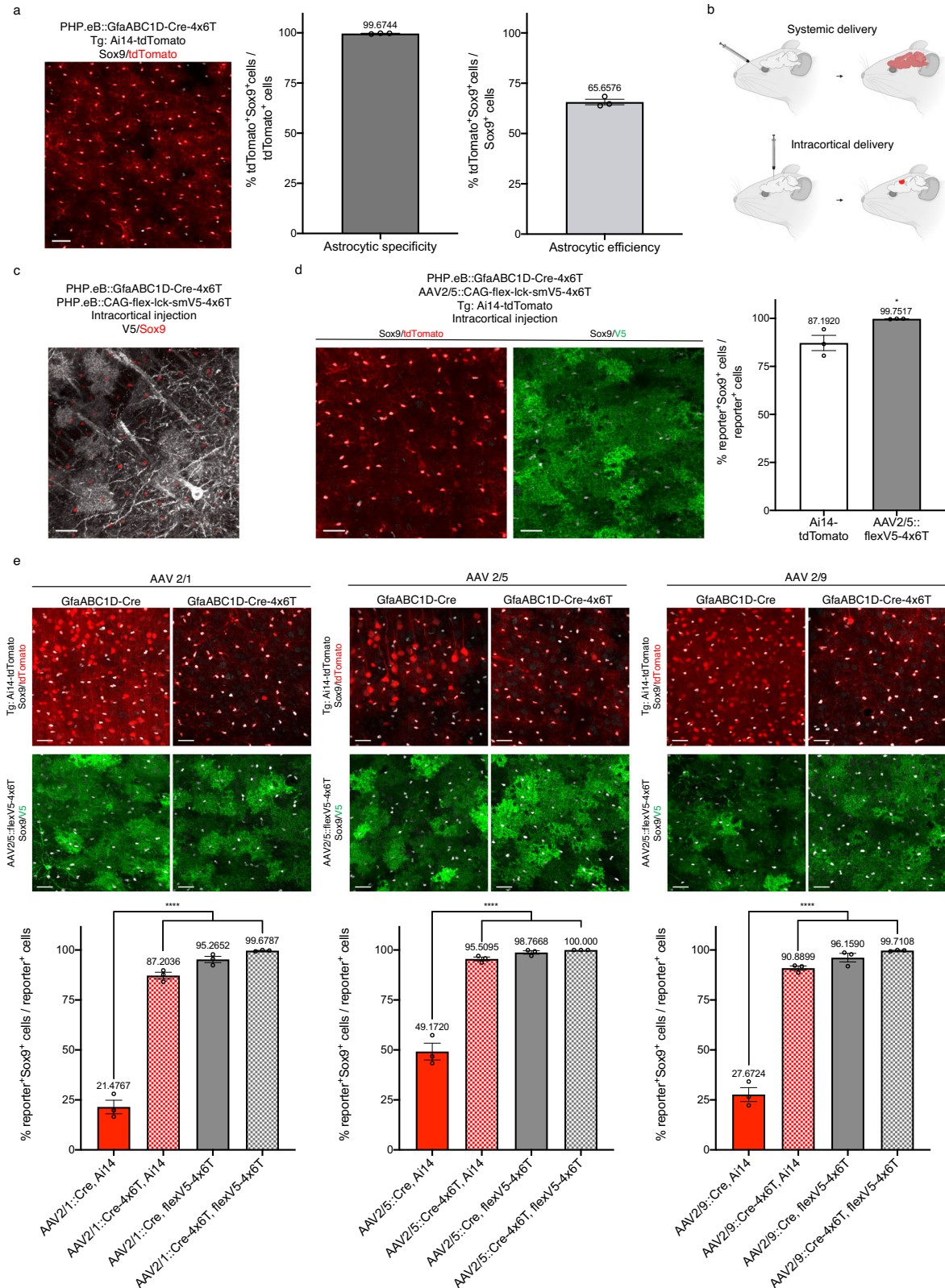

Cre-4x6T: 99.96% ± 0.02 Aldh1l1[+]). Interestingly, both intracortical injections showed higher astrocyte specificity in Ribotag mice than in Ai14-tdTomato mice with similar injections (Fig. 2d). As both transgenic lines are flox-stop-flox reporters, this may reflect differences in how easily Cre can induce recombination in different loci[33]. While we observed significant neuronal expression with AAV2/5::GfaABC1D-Cre (73.97% ± 2.90 Aldh1l1[+]), astrocyte specificity with intracortical

PHP.eB::GfaABC1D-Cre-4x6T was high (98.79% ± 0.32 Aldh1l1[+]), reflecting enhanced astrocyte specificity of the 4x6T miR cassette. We also evaluated GFAP immunoreactivity, a rough metric of astrocyte reactivity[34]; intracortical injection with either serotype induced increases in GFAP, as expected, but systemic injection did not, suggesting that viral transduction alone does not induce overt reactivity (Fig. 5a).

**Fig. 2 | 4x6T cassette increases astrocyte specificity at high titers and across serotypes. a** High-titer ($3 \times 10^{12}$ vg/mouse) systemic delivery of PHP.eB::GfaABC1D-Cre-4x6T in Ai14-tdTomato mice yields 99.67% astrocyte specificity (% of tdTomato$^+$ cells that were Sox9$^+$); the efficiency of transduction (% of Sox9$^+$ cells transduced) was 65.66% of Sox9$^+$ cells in the cortex. Scale bar: 40 μm. **b** Schematic of viral delivery approaches: retroorbital injection leads to lower levels of brain-wide transduction, while direct intracortical injection leads to higher levels of transduction within a narrow region around the injection site. **c** High-titer direct intracortical injection (500 nl of $1 \times 10^{12}$ vg/ml per virus) of PHP.eB::GfaABC1D-Cre-4x6T and PHP.eB::CAG-flex-lck-smV5-4x6T shows high levels of neuronal contamination. Scale bar: 30 μm. **d** High-titer direct intracortical injection of PHP.eB::GfaABC1D-Cre-4x6T and AAV2/5::CAG-flex-lck-smV5-4x6T into Ai14-tdTomato mice shows predominantly astrocytic expression of the tdTomato reporter but more astrocyte-specific expression of the AAV2/5::flexV5-4x6T reporter. Two-tailed unpaired *t* test, *$P = 0.0340$; $t = 3.165$, df=4. Scale bars: 30 μm. **e** High-titer direction intracortical injection of GfaABC1D-Cre-4x6T packaged in different serotypes, co-injected with AAV2/5::CAG-flex-lck-smV5-4x6T into Ai14-tdTomato mice. Serotype impacts astrocyte specificity, with highest specificity with AAV2/5. The presence of the 4x6T on Cre, on the reporter virus, or both increases astrocytic specificity, with the highest specificity seen when both elements of a Cre/reporter system are tagged with 4x6T. Within serotypes: GfaABC1D-Cre vs all 4x6T conditions, one-way ANOVA with Tukey's multiple comparisons test, ****$P < 0.0001$. AAV2/1: $F = 320.2$, df=11. AAV2/5: $F = 124.4$, df=11. AAV2/9: F = 259.0, df=11. Source data are provided as a Source Data file. Scale bars: 30μm. All data presented as mean ± SEM; *n* = 3 mice per condition; all mice 2–5 months old, euthanized 2 weeks post injection.

TRAP transcriptomic data reflect the relative enrichment of ribosomally loaded mRNA vs whole-tissue, or input, mRNA. The effectiveness of the cell-type-specific immunoprecipitation, therefore, can be assessed by evaluating IP-vs-input enrichment of cell-type-specific genes and de-enrichment of genes associated with other cells. We found high astrocytic enrichment in Aldh1l1-CreERT2 mice and both 4x6T cohorts, with lower but still substantial astrocyte enrichment in AAV2/5 samples (Fig. 5b). De-enrichment of non-astrocytic genes was highest in Aldh1l1-CreERT2 mice, lower but substantial in both 4x6T cohorts, and lowest in the AAV2/5 cohort, reflecting the high levels of neuronal contamination in that cohort. Interestingly, endothelial decontamination was comparably robust in Aldh1l1-CreERT2 and both 4x6T cohorts, and weakest in AAV2/5 samples.

Comparing all IP experimental cohorts to Aldh1l1-CreERT2, we found the highest levels of differentially expressed genes (DEGs) in AAV2/5 (Fig. 5c): the greatest molecular difference in gene expression from Aldh1l1-CreERT2, a gold standard of astrocyte mRNA expression, occurs when only the AAV2/5 capsid and minimal GFAP promoter is used to restrict expression to astrocytes. Somewhat unexpectedly, there were few gene expression changes between both 4x6T cohorts (Fig. 5c), although GFAP was upregulated in the intracortical samples compared to systemic injection (Supplementary Data 1), confirming our immunostaining findings. These results suggest that the route of viral administration has a minimal effect on astrocytic gene expression, with intracortical injection inducing only mild reactivity.

Given the lower de-enrichment of non-astrocytic genes in viral cohorts compared to Aldh1l1-CreERT2 (Fig. 5b), we explored which of the viral DEGs were cell-type-specific. We compared these gene lists to cell-type-specific gene signatures, which included the top 1000 most specific genes in astrocytes, endothelial cells, microglia, neurons, and oligodendrocytes, identified using three separate mouse brain cell transcriptomic studies[35]. Most DEGs were not overtly cell-type-specific; neurons made up the largest component of cell-type-specific upregulated genes (Fig. 5d). All of the neuronal upregulated genes in either 4x6T cohort were upregulated in AAV2/5 samples, while 48.8% of neuronal upregulated genes in AAV2/5 samples were not upregulated in either 4x6T cohort (Fig. 5e). The TRAP datasets reflect both any differences in which cell populations are labeled as well as virus-induced changes within the labeled cells: these cell-type-specific enrichment/de-enrichment and gene set analyses demonstrate the higher neuronal contamination of the AAV2/5 samples compared to 4x6T-modified virus, despite the fact that AAV2/5 is a more astrocyte-specific serotype.

To address which transcriptomic changes may occur in astrocytes due to virus transduction itself, we compared these IP datasets with datasets of astrocytes in related pathological settings as well as all Gene Ontology (GO) gene sets. Astrocytes are responsive to environmental perturbations, including pathologic viral infection and direct injury, of which direct intracortical injection is a mild form. The gene changes observed between Aldh1l1-CreERT2 astrocytes and AAV-transduced astrocytes may reflect common transcriptional pathways to those in astrocytes infected with pathologic viruses, to astrocytes after a stab wound injury, or to other biological processes listed in GO. To assess the physiological relevance of the transcriptional changes observed, we used Gene Set Enrichment Analysis[36] (GSEA) to compare our datasets with transcriptomic changes in astrocytes treated with poly I::C to broadly mimic viral stimulation[37], infected with Zika virus[38] or HIV[39], or had undergone a cortical stab wound[40], and all GO gene sets to broadly assess these datasets. GSEA uses a ranked list of all transcriptional changes in the datasets of interest (AAV2/5 and 4x6T cohorts) and assesses where genes in a biologically defined test gene set fall along that ranked list. Overrepresentation of a test gene set at either extreme of the dataset of interest suggests up- or downregulation of that gene set and thus the biological pathway. None of the astrocyte-specific viral or injury datasets were significantly enriched in AAV2/5 or 4x6T-transduced astrocytes (FDR < 0.05, Supplementary Fig. 1), suggesting that the gene changes observed in these cohorts do not reflect common transcriptional pathways induced with pathological viruses or stab wound. Numerous GO gene sets were significantly up- or downregulated (AAV2/5: 305 up, 37 down; 4x6T intracortical: 40 up, 7 down; 4x6T systemic: 192 up, 12 down; Supplementary Data 2); the most significantly upregulated pathways in AAV2/5 reflected neuronal expression, while the most significantly upregulated pathways in both 4x6T cohorts suggest changes in NADH activity (Fig. 5f). The most significantly downregulated pathways in all cohorts are more varied but suggest that AAV transduction in astrocytes may impact metabolism and translation. Together, these results suggest that AAV transduction does not induce overt reactivity in astrocytes but may influence metabolic pathways.

One potential concern in using miRNA targeting sequences is whether those sequences might act as an miRNA sink, by binding endogenous miRNAs and thereby preventing their binding to and regulation of endogenous genes. Therefore, we generated a list of genes in the mouse genome that have shown interaction with the miRNAs in the 4x6T cassette, using the miRNA targeting database TarBase v8[41] (5409 genes). We explored which of these genes showed evidence of differential regulation by 4x6T virus transduction both in input samples, which include neurons and endothelial cells that express the miRs, and in the astrocyte-enriched IP samples. No genes in the input samples were differentially regulated (FDR < 0.05) in either 4x6T cohort compared to Aldh1l1-CreERT2. In IP samples, we evaluated changes that were specific to the presence of the 4x6T cassette: genes that were differentially regulated in either 4x6T cohort but not in the AAV2/5 cohort, reflecting potentially altered expression due to interference with endogenous roles of miRs rather than due to AAV transduction. 88 of 5409 genes fit these criteria (Fig. 5g and Supplementary Data 3), showing very little evidence for unintentional dysregulation of endogenous miRNA function due to the 4x6T cassette.

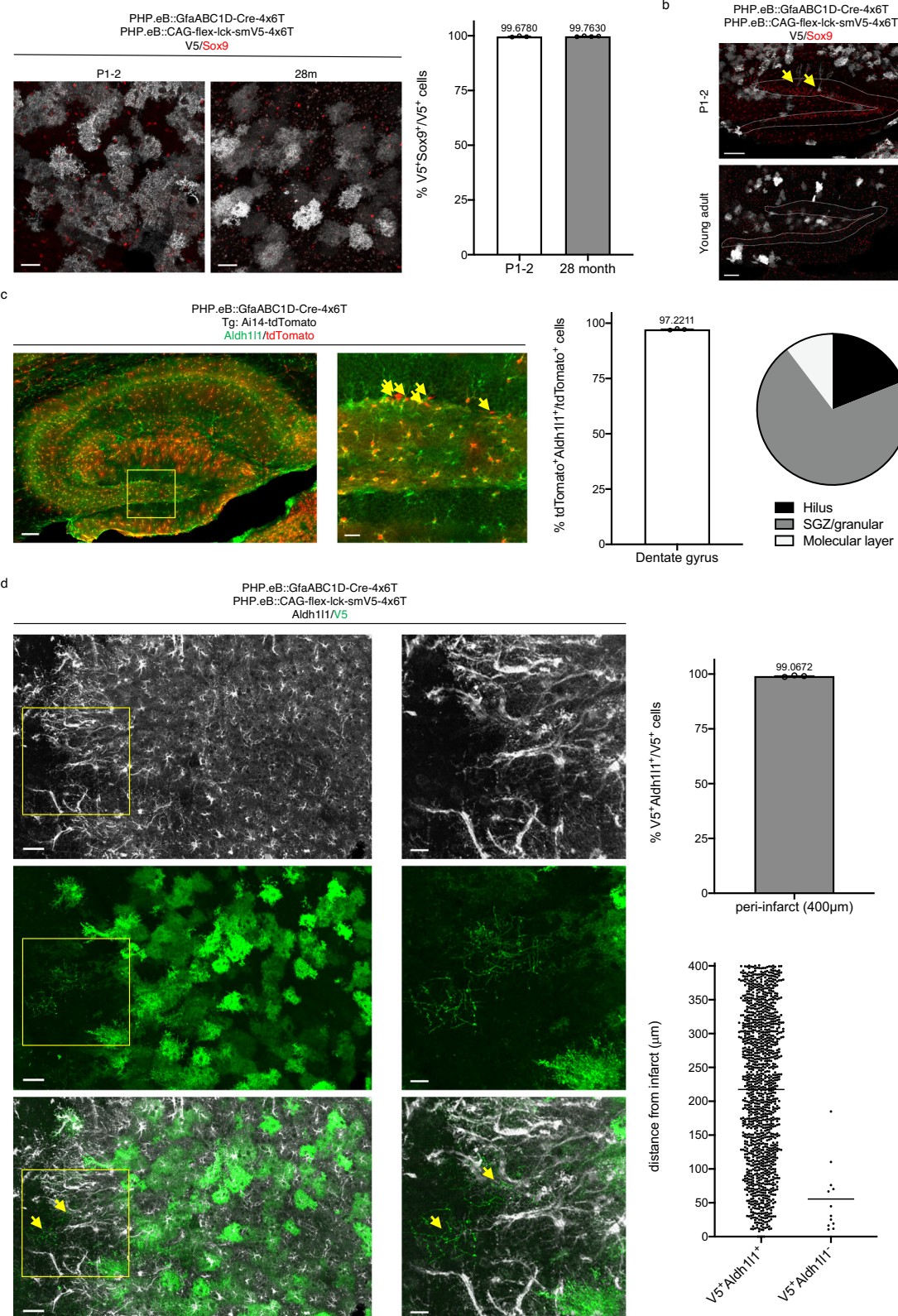

## Broader applicability of the 4x6T cassette in conferring astrocyte specificity

Cre is a particularly sensitive cargo protein to off-target expression, as very low Cre expression can induce genetic recombination. We wondered if the neuronal contamination issues seen with PHP.eB::GfaABC1D-Cre were present more broadly with PHP.eB::GfaABC1D vectors with less sensitive cargos and whether the addition of the 4x6T

might improve specificity. We systemically delivered a spaghetti monster reporter (smMyc; PHP.eB::GfaABC1D-lck-smMyc, $2 \times 10^{11}$ vg/mouse) to assess astrocyte specificity with a reporter protein and found high levels of non-astrocytic contamination (67.78% ± 2.98 Sox9$^+$). In contrast, adding the 4x6T cassette (PHP.eB::GfaABC1D-lck-smMyc-4x6T) significantly improved specificity (99.80% ± 0.05 Sox9$^+$) without affecting efficiency of astrocytic transduction (smMyc: 34.54%

**Fig. 3 | 4x6T cassette confers astrocyte specificity across the lifespan and in reactive astrocytes. a** Systemic injection in mice as young as postnatal days 1–2 (temporal vein injection; $1 \times 10^{10}$ vg/mouse PHP.eB::GfaABC1D-Cre-4x6T + $2 \times 10^{10}$ vg/mouse PHP.eB::CAG-flex-lck-smV5-4x6T; $n = 3$ mice) and as old as 28 months (retroorbital injection; $2 \times 10^{11}$ vg/mouse PHP.eB::GfaABC1D-Cre-4x6T and PHP.eB::CAG-flex-lck-smV5-4x6T; $n = 4$ mice) show highly astrocyte-specific viral expression patterns. Scale bars: 40 μm. **b** Systemic injection in P1-2 pups yields transduced radial glia (yellow arrows) in the dentate gyrus, which is not readily observed in animals injected at 2–5 months; similar observations in four mice/age group. Dotted gray lines, granule cell layer. Scale bars: 100 μm. **c** High-titer retro-orbital PHP.eB::GfaABC1D-Cre-4x6T injection ($3 \times 10^{12}$ vg/mouse) in adult Ai14-tdTomato mice (3–4 months old, $n = 3$ mice) results in high astrocyte specificity in the dentate gyrus. The majority of Aldh1l1⁻tdTomato⁺ cells are found in the SGZ and granular layers (combined totals, 3 mice, 2 sections/mouse; total = 58 cells). Scale bars: left, 100 μm; right, 20 μm. **d** 4x6T cassette maintains high levels of astrocytic specificity after stroke ($n = 3$ mice), although morphologically distinct V5⁺Aldh1l1⁻ can be found near the infarct border. Left, 50 μm scale bar; right, 20 μm scale bar. Yellow box denotes area of higher magnification at right; yellow arrows indicate V5⁺Aldh1l1⁻ cells. V5⁺Aldh1l1⁻ cells: mean distance of 55 μm ± 14.81 from infarct border. Source data are provided as a Source Data file. Bar graphs are presented as mean ± SEM. All mice were euthanized 2 weeks post injection.

± 0.97 Sox9⁺ cells transduced; smMyc-4x6T: 34.79% ± 0.55 Sox9⁺ cells transduced); in non-recombinase vectors, WPRE was included to enhance expression. We observed no difference in the strength of astrocytic Myc expression by immunohistochemistry.

We also generated smV5-4x6T and smFLAG-4x6T vectors; while smV5 showed similar expression patterns to smMyc, we found very sparse transduction with smFLAG when delivered systemically compared to the other smFPs. We found no difference in astrocyte specificity of smV5-4x6T delivered systemically vs direct intracortical injection (99.85% ± 0.01 Sox9⁺, systemic; 99.61% ± 0.17 Sox9⁺, intracortical; Fig. 6b). Further, we found that virtually all astrocytes within the core of the intracortically injected region were transduced; this high efficiency did not come with a trade-off in specificity. These experiments demonstrate that even non-recombinase cargo can show high levels of nonastrocytic expression, which can be improved with the 4x6T cassette.

One advantage of Cre is the development of inducible versions, including both tamoxifen-inducible (ERT2-Cre-ERT2[42]; ERCreER) and light-inducible (iCreV[43]) forms. To maximize the utility of Cre-based 4x6T viruses, we generated 4x6T versions of both ERCreER and iCreV. When we systemically delivered PHP.eB::GfaABC1D-ERCreER-4x6T or PHP.eB::GfaABC1D-iCreV-4x6T with PHP.eB::CAG-flex-lck-smV5, without including a 4x6T cassette on the flex reporter, we found extensive, predominantly neuronal contamination in the absence or presence of the inducing factor (tamoxifen or light, respectively) (Fig. 7a). By including 4x6T cassette on the reporter, however, this contamination was almost fully resolved, generating inducible astrocyte-specific viral manipulation, although we found higher astrocyte specificity with ERCreER (ERCreER, 99.95% ± 0.03 Sox9⁺; iCreV, 97.75% ± 0.52 Sox9⁺; Fig. 7a).

In order to make an astrocyte-specific DNA recombinase that could be used orthogonally with Cre, we generated GfaABC1D-Dre-4x6T and a Dre-dependent reporter, CAG-dDIO-lck-smMyc-4x6T[44]. We co-injected PHP.eB::GfaABC1D-Cre-4x6T, PHP.eB::CAG-flex-lck-smV5-4x6T, PHP.eB::GfaABC1D-Dre-4x6T, and PHP.eB::CAG-dDIO-lck-smMyc-4x6T, and found similar astrocyte specificity with both recombinase/reporter systems (Cre, 99.39% ± 0.15 Sox9⁺; Dre, 99.34% ± 0.23 Sox9⁺; Fig. 7b). At non-saturating titers there is some, but not complete, overlap in Dre- and Cre-labeled cells; these viruses, therefore, can be used to label neighboring astrocytes with different combinations of reporters to evaluate astrocytic tiling (Fig. 7b).

The 4x6T vectors presented here—GfaABC1D-Cre, ERCreER, iCreV, Dre, and smFPs; conditional reporters CAG-flex-lck-smV5 and dDIO-lck-smMyc—are ready-to-package AAV plasmids for astrocyte-specific manipulation and morphologic analysis (Table 1). In order to facilitate the addition of 4x6T to other cargo as needed, we further generated empty vectors in which the cargo protein was replaced with a multiple cloning sequence (MCS). We generated three such vectors: GfaABC1D-MCS--4x6T, CAG-flex-MCS--4x6T, and CAG-dDIO-MCS--4x6T (Fig. 7c and Table 1). Together, these vectors form a toolbox of astrocyte-specific AAV plasmids to facilitate astrocytic viral manipulation across a wide range of experimental conditions.

## Discussion

Here, we present the development of a miRNA targeting cassette comprised of four copies of each of six miR targeting sequences (4x6T). This multiplexed cassette de-targets both neurons and endothelial cells, demonstrating the flexible use of miR targeting in improving viral specificity by decreasing multiple off-target populations simultaneously. The 4x6T cassette (i) confers a high level of astrocyte specificity on AAV vectors across all serotypes and cargos tested, (ii) exhibits specificity across the mouse lifespan and in injury conditions, and (iii) maintains astrocyte-specific expression patterns for at least 6 months post injection. We have used the 4x6T cassette to create a toolbox of astrocytic AAV vectors that will enable cleaner, more flexible astrocytic genetic manipulation across a wide range of experimental conditions.

While we focused this analysis on the cerebral cortex, we found that systemic delivery of PHP.eB::GfaABC1D viruses transduces astrocytes across the CNS, with the notable exception of white matter, where we found very low transduction. White matter astrocytes were transduced with intracortical injection of different serotypes, suggesting this may be an issue of serotype and/or delivery route. Although the 4x6T cassette dramatically improved astrocyte specificity of all tested serotypes, given the level of neuronal contamination upon intracortical high-titer injection of PHP.eB::CAG-flex-lck-smV5-4x6T, it appears that high titer combined with a generic promoter can overcome miR regulation. Therefore, for high-titer direct injections of flex cassettes with a strong generic promoter, we recommend AAV2/5 for the highest possible astrocytic specificity. We also found slightly lower astrocytic specificity in the hippocampus, particularly the dentate gyrus; however, even when the high-titer virus was delivered to adult mice the level of off-target labeling remained quite low, suggesting that there may be some de-targeting of progenitor cells by the 4x6T cassette. This would be consistent with the high miR-124 levels found in progenitor cells[45].

This modification to improve astrocyte specificity is compatible with different capsids, makes it possible to take advantage of the growing capabilities of designer AAV capsids; here, we used the systemically deliverable PHP.eB capsid, but 4x6T could be used with serotypes with other capabilities, such as delivery to the peripheral nervous system (PHP.S[4]). The 4x6T cassette is also promoter-independent and could be used with different promoters or enhancers to potentially restrict expression to specific astrocytic sub-populations. Given the extensive and growing evidence of astrocyte heterogeneity[46–49], the ability to rapidly, specifically genetically manipulate distinct astrocytic subpopulations would be extremely powerful; the 4x6T cassette makes that more feasible and opens up new avenues for astrocyte-specific gene therapy.

The repeating nature of the cassette makes de novo synthesis challenging and complicates PCR-based cloning. In contrast, the empty vectors presented here facilitate rapid, easy generation of 4x6T vectors with any number of cargo. These vectors, combined with the reporters and recombinases generated here, provide a wide range of options to enable highly specific, rapid genetic manipulation of astrocytes.

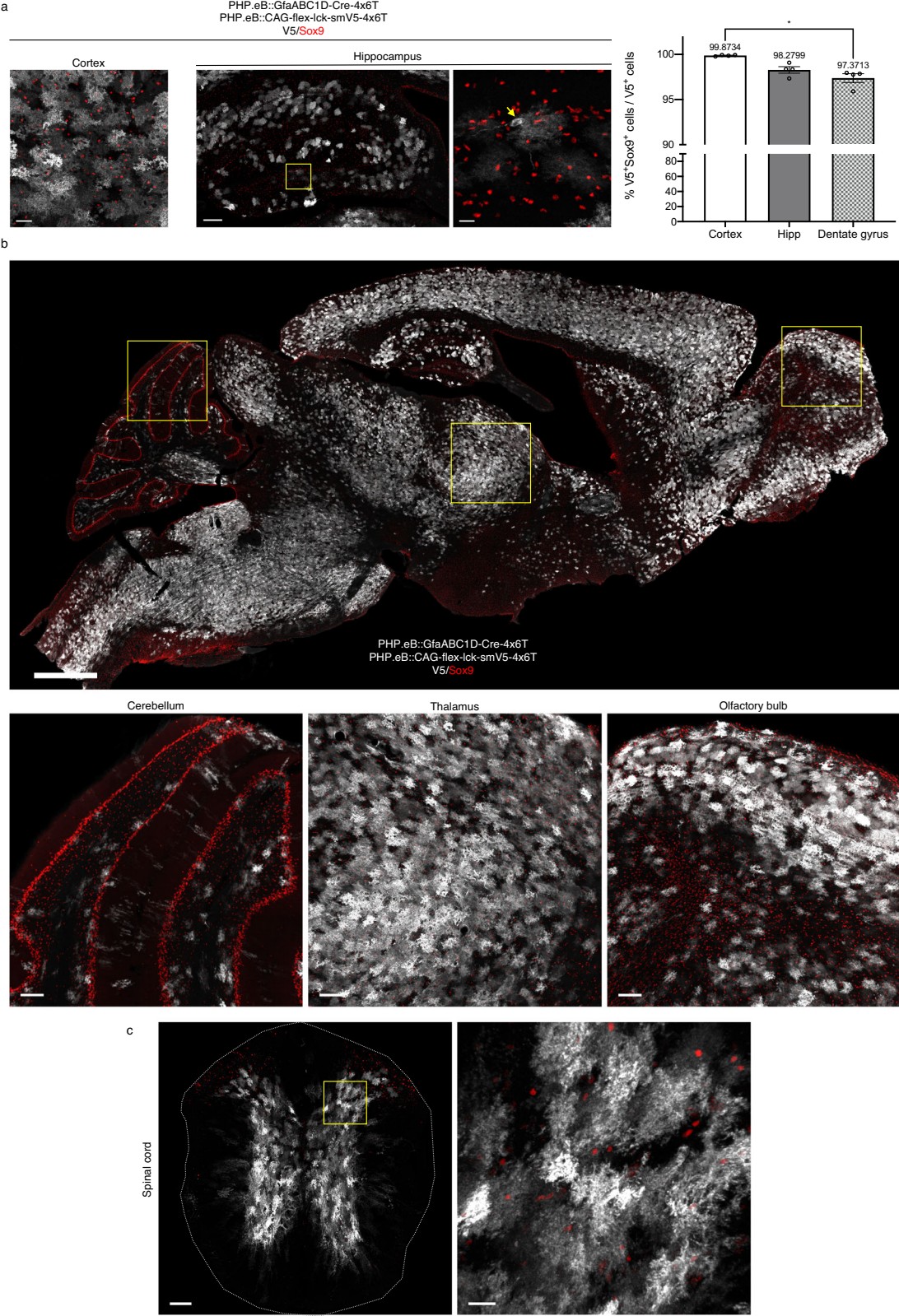

## Methods

### Generation of miR-containing plasmids

Plasmids were constructed using Gibson assembly[50]. Viral backbones were generated by restriction enzyme digest. Inserts were generated by polymerase chain reaction (PCR); primers were designed using NEBuilder (New England BioLabs). All plasmids were confirmed by whole-plasmid sequencing (Primordium Labs).

AAV::GfaABC1D plasmids: AAV::GfaABC1D plasmids were based on backbone plasmid AAV-pmSyn1-EBFP-Cre[51]. AAV-pmSyn1-EBFP-Cre was a gift from Hongkui Zeng (Addgene plasmid # 51507; http://n2t.net/addgene:51507; RRID:Addgene_51507). Two copies of single miR-T sequences were added in primers, split between the 3′ end of the Cre insert and the 5′ end of hGH polyA regulatory sequence insert. The 4x6T cassette was constructed in multiple steps, which was necessary

**Fig. 4 | Astrocyte specificity of 4x6T cassette is preserved for long time periods and across CNS regions. a** Astrocyte specificity 6 months after injection (retro-orbital injection, $5 \times 10^{11}$ vg/mouse PHP.eB::GfaABC1D-Cre-4x6T; $5 \times 10^{11}$ vg/mouse PHP.eB::CAG-flex-lck-smV5-4x6T) into young adult (2–5-month-old) mice is preserved in the cortex (scale bar: 40 μm) and to a slightly lesser degree in the hippocampus. Hippocampus, left: entire structure (scale bar: 150 μm); yellow box shows the region of higher magnification on the right (scale bar: 20 μm). Yellow arrow: example of neuron in the dentate gyrus. Astrocyte specificity is higher in the cortex than the dentate gyrus sub-region (*$P = 0.0134$; Kruskal–Wallis test, Dunn's multiple comparisons test, $P = 0.0024$, Kruskal–Wallis statistic 8.346). Mean ± SEM; $n = 4$ mice per brain region. Source data are provided as a Source Data file. **b** Sagittal section of mouse brain 6 months after virus injection (scale bar: 1 cm), with higher magnification examples of the cerebellum, thalamus, and olfactory bulb (scale bars: 100 μm), showing high levels of colocalization of V5 with Sox9. **c** Coronal section of mouse spinal cord 6 months after virus injection (scale bar: 100 μm); yellow box denotes area of higher magnification section on the right (scale bar: 20 μm), showing high V5/Sox9 colocalization; similar observations in four mice in each CNS region.

due to complexities in synthesizing high numbers of repeating DNA sequences. Two copies of each of the 5 neuronal miR-T (2x5T) were synthesized as a single gene product (gBlock, Integrated DNA Technologies) and added to a Cre plasmid. This 2x5T plasmid was then used as part of the backbone to add an additional two copies of each neuronal miR via PCR insert, to which four copies of miR-126T were added in primers. Additional restriction enzyme sites were added in primers to facilitate downstream cloning. AAV::GfaABC1D-Cre-4x6T plasmid was used as a backbone to generate other GfaABC1D-(cargo)-4x6T plasmids. ERCreER: pCAG-ERT2CreERT2 was a gift from Connie Cepko (Addgene plasmid # 13777; http://n2t.net/addgene:13777; RRID:Addgene_13777)[42]. iCreV: pAAV-EF1a-iCreV was a gift from Hongkui Zeng (Addgene plasmid # 140135; http://n2t.net/addgene:140135; RRID:Addgene_140135)[43]. Dre: AAV phSyn1(S)-DreO-bGHpA was a gift from Hongkui Zeng (Addgene plasmid # 50363; http://n2t.net/addgene:50363; RRID:Addgene_50363)[51]. In GfaABC1D-smFP-4x6T-WPRE plasmids, WPRE was added to boost spaghetti monster fusion protein (smFP) reporter expression. pCAG_smFP V5 (Addgene plasmid # 59758; http://n2t.net/addgene:59758; RRID:Addgene_59758), pCAG_smFP FLAG (Addgene plasmid # 59756; http://n2t.net/addgene:59756; RRID:Addgene_59756), and pCAG_smFP Myc (Addgene plasmid # 59757; http://n2t.net/addgene:59757; RRID:Addgene_59757) were gifts from Loren Looger[23].

AAV::CAG plasmids: AAV::CAG plasmids were based on backbone plasmid AAV-FLEX-GFP. pAAV-FLEX-GFP was a gift from Edward Boyden (Addgene plasmid # 28304; http://n2t.net/addgene:28304; RRID:Addgene_28304). Reverse-complement inserts for smFPs and the 4x6T cassette were generated via PCR; given the repeating nature of the 4x6T cassette, multiple 4x6T PCR products were generated. These products were separated using 3% low-melting-temperature agarose gels to ensure isolation and purification of the proper molecular weight product. Flex construct lox sites were replaced with rox sites to generate Dre-dependent dDIO constructs, using pAAV-Ef1a-dDIO hChR2(H134R)-EYFP; pAAV-Ef1a-dDIO hChR2(H134R)-EYFP was a gift from Karl Deisseroth (Addgene plasmid # 55640; http://n2t.net/addgene:55640; RRID:Addgene_55640)[44].

Empty vector plasmids: Existing GfaABC1D-smFP-4x6T-WPRE and CAG-flex/dDIO-smFP-4x6T plasmids were used as backbone plasmids, with cargo removed via restriction enzyme digest. Multiple cloning sequences (MCS) comprised of restriction enzyme sites that were absent from the backbone plasmid were generated via gene synthesis (Integrated DNA Technologies) and added via Gibson assembly.

The following plasmids are available on Addgene: AAV-GfaABC1D-Cre-4x6T (196410), AAV-GfaABC1D-ERCreER-4x6T (196411), AAV-GfaABC1D-iCreV-4x6T (196412), AAV-GfaABC1D-DreO-4x6T (196413), AAV-GfaABC1D-lck-smFLAG-4x6T-WPRE (196414), AAV-GfaABC1D-lck-smMyc-4x6T-WPRE (196415), AAV-GfaABC1D-lck-smV5-4x6T-WPRE (196416), AAV-GfaABC1D-MCS--4x6T-WPRE (196417), AAV-CAG-flex-GFP-4x6T (196418), AAV-CAG-flex-lck-smV5-4x6T (196419), AAV-CAG-flex-MCS--4x6T (196420), AAV-CAG-dDIO-lck-smMyc-4x6T (196421), AAV-CAG-dDIO-MCS--4x6T (196422), AAV-CAG-flex-lck-smV5 (196423).

## AAV packaging and titration

All AAVs used in this study were produced in the Carmichael lab, in accordance with the US National Institutes of Health Guidelines for Research Involving Recombinant DNA Molecules and the University of California Los Angeles Institutional Biosafety Committee, with the exception of PHP.eB::GFAP-Cre. pAAV.GFAP.Cre.WPRE.hGH was a gift from James M. Wilson (Addgene viral prep # 105550-PHPeB; http://n2t.net/addgene:105550; RRID:Addgene_105550).

Rep/cap plasmids: PHP.eB: pUCmini-iCAP-PHP.eB was a gift from Viviana Gradinaru (Addgene plasmid # 103005; http://n2t.net/addgene:103005; RRID:Addgene_103005)[4]. AAV2/1: pAAV2/1 was a gift from James M. Wilson (Addgene plasmid # 112862; http://n2t.net/addgene:112862; RRID:Addgene_112862). AAV2/5: pAAV2/5 was a gift from Melina Fan (Addgene plasmid # 104964; http://n2t.net/addgene:104964; RRID:Addgene_104964). AAV2/9: pAAV2/9n was a gift from James M. Wilson (Addgene plasmid # 112865; http://n2t.net/addgene:112865; RRID:Addgene_112865).

AAV packaging was performed by transient transfection of HEK293 cells (ATCC #CRL 3216) followed by iodixanol gradient purification[52]. In total, 5–10 150-mm dishes of HEK293 cells were transfected with pAAV, capsid, and helper plasmids at a ratio of 1:4:2 using polyethylenimine transfection. The media was changed the following day, then media was collected at 3 and 5 days post-transfection as well as cells at 5 days. Cells were pelleted and resuspended in 500 mM NaCl, 40 mM Tris, 10 mM MgCl$_2$, supplemented with salt-activated nuclease (100 U/ml, ArcticZymes), and incubated for 1 h at 37 °C. Viral particles were precipitated from media via incubation with 8% PEG8000 at 4 °C for 2 h followed by 30 min centrifugation at $4000 \times g$ for 30 min at 4 °C; the resulting pellet was also resuspended in salt-activated nuclease buffer and incubated with the cell lysate for 30 min at 37 °C. Iodixanol density gradients were poured; from the top, 15%, 25%, 40%, and 60% steps; 25% and 60% steps included ~1% phenol red for visualization. Lysates were then pelleted at $3000 \times g$ for 15 min and loaded onto iodixanol gradients. Gradients were then spun at $350,000 \times g$ on an ultracentrifuge for 2 h 25 min at 18 °C.

To exchange the remaining iodixanol from purified virus, Amicon filtration columns were used (Millipore, 100 K MW). Filtration columns were covered with 15 ml DPBS + 0.1% Pluronic F-68 for 10 min; solution was discarded and replaced with 15 ml DPBS + 0.01% Pluronic F-68. Columns were centrifuged at $3000 \times g$ for 3 min; flow-through was discarded. The viral layer was removed using a syringe with an 18 G needle and loaded into the filtration column in combination with 10 ml DPBS + 0.001% Pluronic F-68 through a 0.22-μm syringe filter (Millipore). Buffer was then exchanged four times through the filtration column with $3000 \times g$ centrifugation using DPBS + 0.001% Pluronic F-68. A final volume of 200–800 μl of virus after the final buffer exchange was recovered from the column, filtered through a 0.22-μm syringe filter, and stored at 4 °C.

Virus was titered using qPCR[52]. Briefly, virus was incubated with DNAse I (1U/ml, ThermoFisher) for 1 h at 37 °C to remove unpackaged plasmid DNA; DNAse I was then inactivated by the addition of EDTA and incubation at 70 °C for 10 min. The viral capsid was then digested with Proteinase K (New England Biolabs) for 2 h at 50 °C; Proteinase K was inactivated via 95 °C incubation for 10 min. Viral samples were then run on a qPCR plate alongside DNA standards using SYBR green master mix (Roche). qPCR primers used were based on the pAAV plasmid: WPRE Fwd, GGCTGTTGGGCACTGACAAT; Rev, CCGAAG

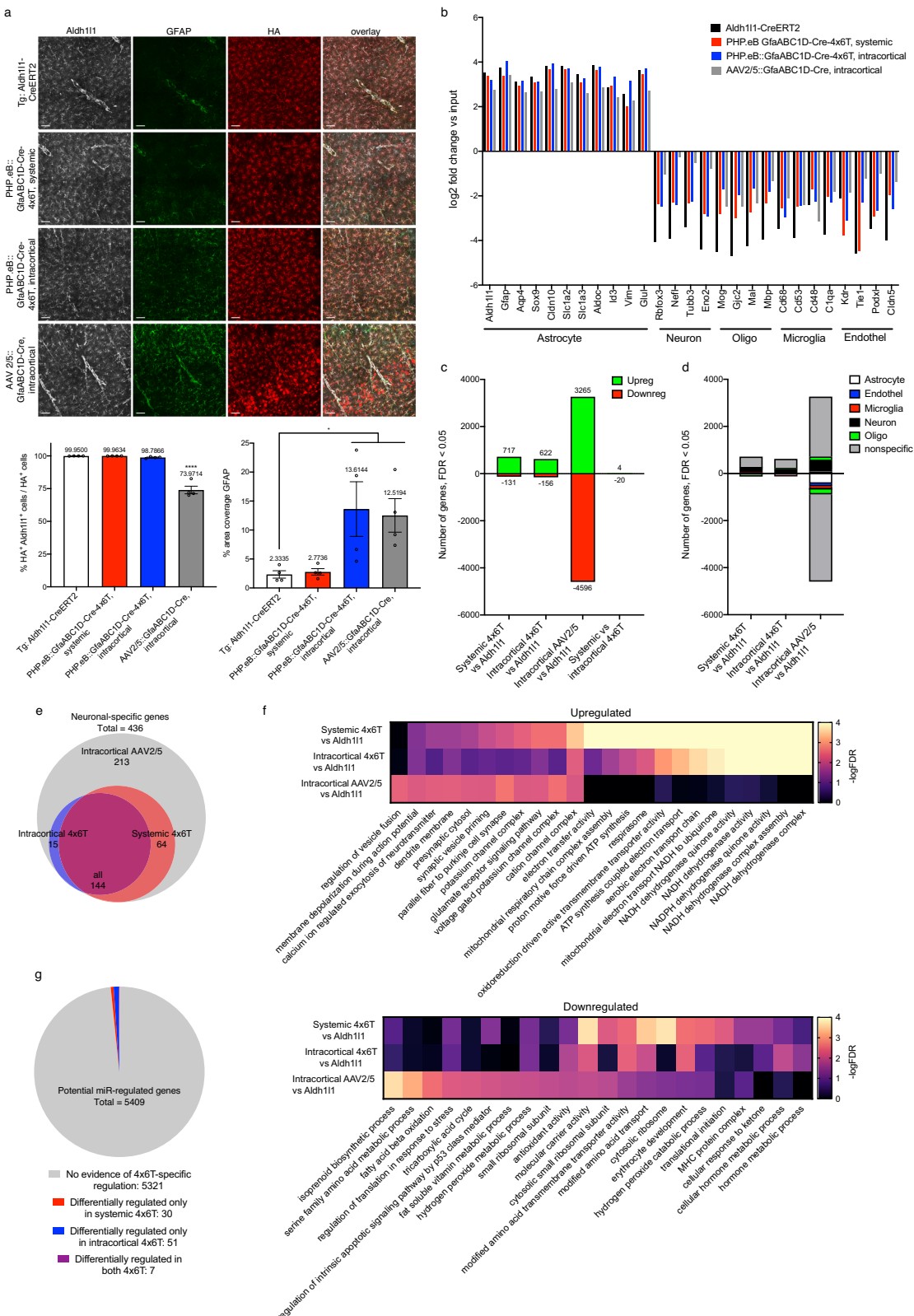

GGACGTAGCAGAAG[52]. hGH primers: Fwd: CCTGGGTTCAAGCGATT CT; Rev: CAGCCTGGCCAATATGGT.

## Animals

Animal procedures were performed in accordance with the US National Institutes of Health Animal Protection Guidelines and the University of California Los Angeles Chancellor's Animal Research Committee, protocol 2000-159. All mice were housed in a facility with 12 h–12 h light–dark cycle, controlled temperature and humidity, and had ad libitum access to food and water. Young adult (8–10 weeks) C57Bl/6J mice were purchased from Jackson Labs (000664) and injected between 2 and 5 months of age; aged C57Bl/6J mice (18–20 months) were obtained from the National Institute on Aging Aged Rodent Colony and injected at 28 months of age. Conditional

**Fig. 5 | Transcriptional analysis of astrocytes with viral transduction with and without 4x6T cassette (TRAP). a** Immunohistochemical analysis of RiboTag⁺ cells (hemagglutinin HA⁺ ribosomal tag), colocalized with astrocytic marker Aldh1l1 and astrocytic reactivity marker GFAP. Astrocytic specificity is high in Aldh1l1-CreERT2 transgenic mice and systemic delivery of PHP.eB::GfaABC1D-Cre-4x6T with no overt evidence of astrocyte reactivity (GFAP% coverage). Specificity remains high with intracortical delivery of PHP.eB::GfaABC1D-Cre-4x6T and decreases with intracortical delivery of AAV2/5::GfaABC1D-Cre; this route of viral delivery shows some evidence of astrocyte reactivity by GFAP immunoreactivity. Scale bars: 50 µm. Mean ± SEM; $n = 4$ mice per cohort. Astrocyte specificity, HA⁺Aldh1l1⁺/HA⁺: one-way ANOVA, Holm–Sidak's multiple comparisons test, $P < 0.0001$, $F = 77.01$, df=15; Aldh1l1-CreERT2 vs AAV2/5::GfaABC1D-Cre ****$P < 0.0001$. GFAP % area coverage: Aldh1l1-CreERT2 = 2.33% coverage ± 0.64; PHP.eB::GfaABC1D-Cre-4x6T systemic, 2.77% coverage ± 0.55; PHP.eB::GfaABC1D-Cre-4x6T intracortical, 13.61% coverage ± 4.71; AAV2/5::GfaABC1D-Cre, 12.52% coverage ± 2.90; one-way ANOVA, Holm–Sidak's multiple comparisons test, $P = 0.0011$, $F = 10.54$, df=15; Aldh1l1-CreERT2 vs PHP.eB::GfaABC1D-Cre-4x6T intracortical *$P = 0.0431$; Aldh1l1-CreERT2 vs AAV2/5::GfaABC1D-Cre *$P = 0.0481$. Source data are provided as a Source Data file. **b** Relative levels of enrichment and de-enrichment of canonical genes for astrocytes, neurons, oligodendrocytes, microglia, and endothelial cells in IP-vs-input samples. **c** Differentially expressed genes in IP samples: different viral cohorts

vs Aldh1l1-CreERT2 IP samples, and systemic vs cortical 4x6T samples. FDR < 0.05, average FPKM across all IP samples >1. **d** Cell-type specificity of differentially expressed genes in IP samples: genes in (**c**) that are represented in the top 1000 specific genes for astrocytes, endothelial cells, microglia, neurons, or oligodendrocytes. **e** Venn diagram showing the overlap of which neuron-specific genes from (**d**) are upregulated in different viral cohort IP samples vs Aldh1l1-CreERT2 samples. **f** Top ten most significantly upregulated and downregulated Gene Ontology gene sets (FDR < 0.05) in ranked IP transcriptomes of each of the viral cohorts, calculated with Gene Set Enrichment Analysis; some gene sets overlapped, particularly for 4x6T cohorts, and only seven gene sets passed FDR < 0.05 for intracortical 4x6T. In cases where the FDR was 0, scores were reassigned as 0.0001 to visualize the relative -logFDR. **g** Numbers of genes potentially regulated by miRNAs that comprise the 4x6T cassette (5409 total) that show evidence of 4x6T-specific regulation in IP samples (FDR < 0.05 in either 4x6T cohort vs Aldh1l1-CreERT2 cohort; not differentially regulated in AAV2/5 cohort vs Aldh1l1-CreERT2 cohort). Note: none of the 5409 genes show evidence of 4x6T-specific regulation in input samples. $n = 4$ mice per cohort; 2–4 months old, euthanized 2 weeks after final tamoxifen administration and 18 days postvirus injection. Titer: retroorbital PHP.eB::GfaABC1D-Cre-4x6T, $1 \times 10^{12}$ vg/mouse; intracortical, PHP.eB::GfaABC1D-Cre-4x6T or AAV2/5::GfaABC1D-Cre: 500 nl of $1 \times 10^{12}$ vg/ml in each of three sites.

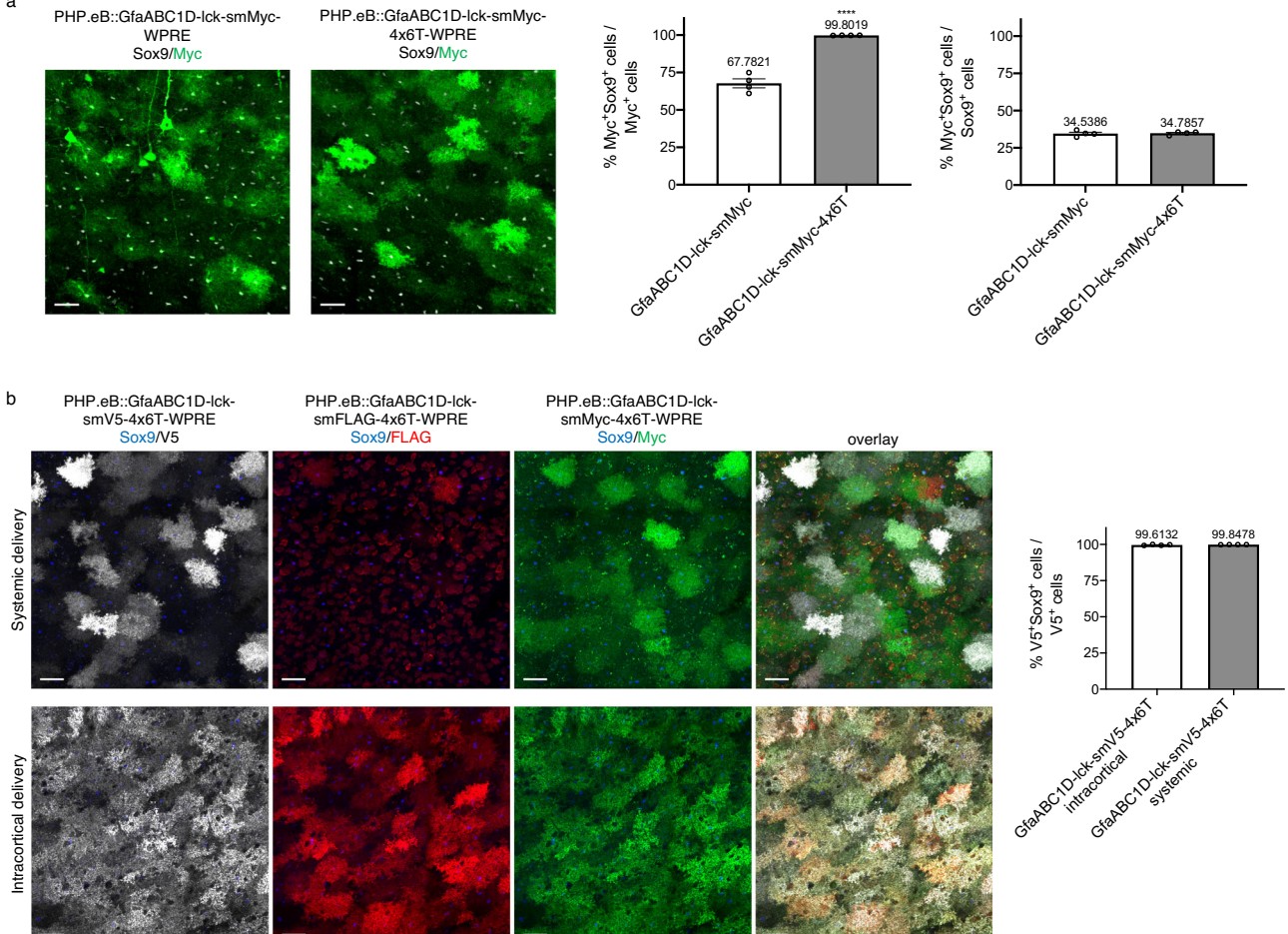

**Fig. 6 | Non-recombinase astrocyte specificity with and without the 4x6T cassette. a** PHP.eB::GfaABC1D driving a spaghetti monster reporter (smMyc) shows high levels of non-astrocytic contamination, while adding a 4x6T cassette significantly increases astrocyte specificity ($n = 4$ mice per cohort, two-tailed $t$ test, ****$P < 0.0001$, $t = 10.73$, df=6). The addition of the 4x6T cassette does not affect astrocyte transduction efficiency (two-tailed $t$ test, $P = 0.8314$, $t = 0.2224$, df=6). **b** Astrocytic specificity remains high with spaghetti monster reporter smV5 constructs delivered either via direct intracortical injection or systemically ($n = 4$ mice

per cohort, two-tailed $t$ test, $P = 0.2099$, $t = 1.404$, df=6). Adding the 4x6T cassette to other smFPs (FLAG, Myc) also show high astrocyte specificity, although PHP.eB::GfaABC1D-smFLAG-4x6T shows surprisingly low transduction efficiency when delivered systemically. Source data are provided as a Source Data file. All data presented as mean ± SEM. Scale bars: 40 µm. Retroorbital virus delivery: each virus $2 \times 10^{11}$ vg/mouse; intracortical: 500 nl of $1 \times 10^{12}$ vg/ml in each of two sites; 2–4-month-old mice, euthanized 2 weeks post injection. Note: all GfaABC1D-smFP constructs included a WPRE regulatory element to boost transgene expression.

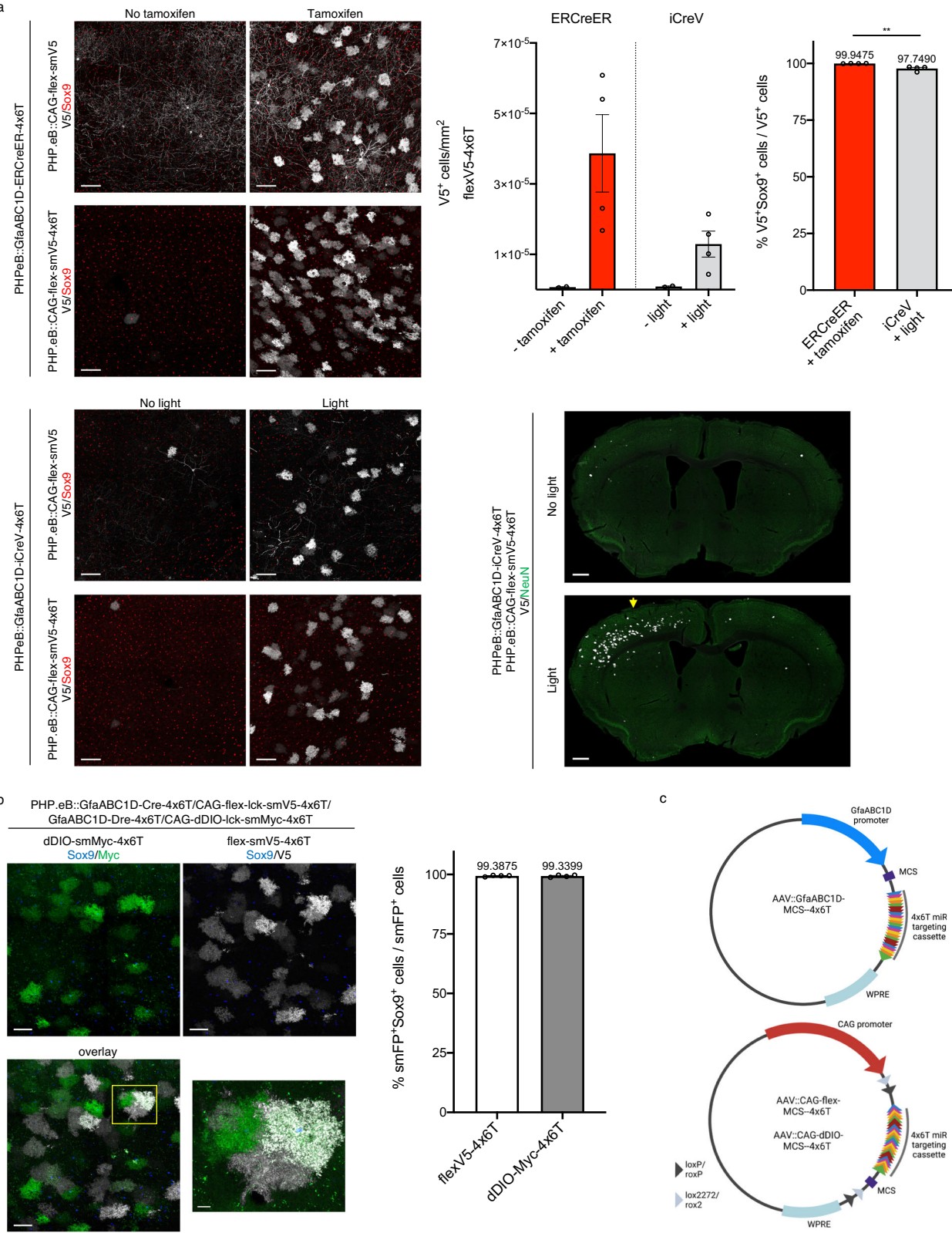

### AAV delivery, Cre induction, and stroke

Systemic AAV delivery in adult mice was performed via retroorbital injection[52]. Briefly, animals were anesthetized via isoflurane and the virus was delivered via a syringe with a 27 G needle inserted into the medial canthus at a 45-degree angle behind the eye. Virus was diluted in sterile saline to a standard volume of 50 μl per mouse; as needed based on titer, volume was increased up to 100 μl. Systemic AAV

Cre-dependent mouse lines used were Ai14 (Rosa-CAG-LSL-tdTomato-WPRE)[19] as a general reporter and RiboTag (loxP-STOP-loxP-Rpl22-HA)[31] for TRAP transcriptomic analysis. RiboTag mice were crossed with Aldh1l1-Cre/ERT2[32] to generate an astrocyte-specific TRAP line; Cre-negative, RiboTag-positive littermates were used for viral delivery TRAP. Three to five mice were used per condition for each experiment and included both male and female mice.

**Fig. 7 | Astrocyte specificity of the 4x6T cassette with inducible and alternative recombinases. a** Tamoxifen- (ERCreER) and light- (iCreV) inducible forms of Cre show high levels of neuronal background without induction when co-injected with a flexV5 reporter with no 4x6T cassette (PHP.eB::CAG-flex-smV5) but much less background when co-injected with a flexV5-4x6T reporter (PHP.eB::CAG-flex-smV5-4x6T). Scale bars: 100 μm. Astrocyte specificity with a flexV5-4x6T reporter is high with both inducible forms of Cre, but higher in ERCreER (two-tailed unpaired *t* test, **$P = 0.0055$, $t = 4.233$, df=6; $n = 4$ mice with tamoxifen or light; two mice with no tamoxifen or no light). Light induction of iCreV shows expression across a broad area of cortex ipsilateral to light placement and limited contralateral expression. Light placement: yellow arrow. Scale bar: 500 μm. All: 2–5-month-old mice, euthanized 2 weeks after Cre induction, 4–4.5 weeks after retroorbital virus injection. Titers: ERCreER-4x6T, iCreV-4x6T: $5 \times 10^{11}$ vg/mouse + CAG-flex-smV5-4x6T: $5 \times 10^{11}$ vg/mouse. **b** PHP.eB::GfaABC1D-Dre-4x6T and Dre-dependent reporter PHP.eB::CAG-dDIO-smMyc-4x6T can be used orthogonally with Cre/flex viral systems and show similar levels of astrocytic specificity (scale bar: 50 μm); yellow box shows the region of higher magnification on the right (scale bar: 10 μm); $n = 4$ mice per recombinase, 2–3-month-old mice, euthanized 2 weeks after retroorbital virus delivery. Titers: Dre-4x6T: $2 \times 10^{11}$ vg/mouse; Cre-4x6T, dDIO-smMyc-4x6T, flex-smV5-4x6T: $1 \times 10^{11}$ vg/mouse. Source data are provided as a Source Data file. All data are presented as mean ± SEM. **c** Schematic diagrams of empty vectors with multiple cloning sequences (MCS) for insertion of other cargo: GfaABC1D-MCS-4x6T for non-recombinase-dependent expression; CAG-flex-MCS-4x6T for Cre-dependent expression; and CAG-dDIO-MCS-4x6T for Dre-dependent expression. Note the presence of a woodchuck hepatitis virus post-transcriptional regulatory element (WPRE); while we omitted this element in the case of recombinase vectors, where high levels of transgene expression were neither wanted nor needed, we have included it in these more general vectors. More detail on transgene components and the full sequences can be found on Addgene.

delivery in P1-2 pups was performed via a temporal vein in a volume of 25 μl per animal, diluted in sterile saline, and loaded into an insulin syringe with a 30-G needle. Pups were anesthetized by placing them on wet ice, then placed under a dissection microscope. The syringe was inserted into the temporal vein and virus was injected[53].

Systemic virus was delivered at $5 \times 10^{11}$ vg/mouse per virus, with the following exemptions: GFAP-Cre, $5 \times 10^{10}$ vg/mouse (Fig. 1a); high titer, $3 \times 10^{12}$ vg/mouse (Fig. 2a); P1-2 pups, PHP.eB::GfaABC1D-Cre-4x6T + $2 \times 10^{10}$ vg/mouse PHP.eB::CAG-flex-lck-smV5-4x6T (Fig. 3a, b); 28-month-old mice, $2 \times 10^{11}$ vg/mouse (Fig. 3a); systemic PHP.eB::GfaABC1D-Cre-4x6T in Ribotag mice, $1 \times 10^{12}$ vg/mouse (Fig. 5a); PHP.eB:::GfaABC1D-lck-smMyc-WPRE, PHP.eB::GfaABC1D-lck-smMyc-4x6T-WPRE, PHP.eB::GfaABC1D-lck-smFLAG-4x6T-WPRE, PHP.eB::GfaABC1D-lck-smV5-4x6T-WPRE, $2 \times 10^{11}$ vg/mouse (Fig. 6a, b); PHP.eB::GfaABC1D-Cre-4x6T/CAG-flex-lck-smV5-4x6T/CAG-dDIO-lck-smMyc-4x6T: $1 \times 10^{11}$ vg/mouse per virus, GfaABC1D-Dre-4x6T: $2 \times 10^{11}$ vg/mouse (Fig. 7b).

Intracortical virus was delivered under isoflurane anesthesia in a stereotactic apparatus via pulled glass micropipette. 500 nl of virus diluted in sterile saline was delivered to each of two locations in the cortex (A/P: 0.0 mm, M/L: 2.0 mm, D/V: −0.75 mm; A/P: −1.2 mm, M/L: 2.0 mm, D/V: −0.65 mm). For RiboTag intracortical cohorts, a third injection was performed in the contralateral cortex (A/P: −0.6 mm, M/L: −2.0 mm, D/V: −0.75 mm); ipsilateral tissue was used for ribosome immunoprecipitation, while contralateral tissue was used for immunohistochemistry. The pipette was left in situ for 3 min to allow proper virus diffusion. Intracortical virus was delivered at $1 \times 10^{12}$ vg/ml in all experiments.

Cre induction: Tamoxifen was administered at 100 mg/kg in corn oil i.p. for 5 days, beginning 2 weeks after virus injection (PHP.eB::GfaABC1D-ERCreER-4x6T) or on the day of virus administration in other RiboTag experimental cohorts (Aldh1l1-CreERT2 transgenic mice). Light induction: 2 weeks after virus administration, (PHP.eB::GfaABC1D-iCreV-4x6T) mice were placed in a stereotactic apparatus under isoflurane anesthesia. A cold light source (KL1500 LCD; Carl Zeiss MicroImaging) attached to a 40x objective was positioned at the skull (A/P: 0.0 mm, M/L: 2.0 mm) and illuminated at maximum illumination for 40 min.

Middle cerebral artery occlusion: The distal middle cerebral artery (MCA) was exposed by transection of the temporal muscle and craniotomy directly over a branch of the MCA. Distal MCA occlusion was produced via electrocoagulation and permanent occlusion of the distal branch of the middle cerebral artery[54], followed by 15 min bilateral jugular vein clamp. Rectal temperature was maintained at 37 °C ± 0.5 °C throughout surgery.

## Tissue processing and immunohistochemistry
Animals were euthanized 2 weeks post-injection or post-Cre-induction; for long-term expression, animals were euthanized 6 months post-injection. Animals were exposed to terminal levels of isoflurane and transcardially perfused with PBS followed by cold 4% paraformaldehyde in PBS (PFA). The brain was removed, postfixed in 4% PFA for 3–6 h, and cryoprotected in 30% sucrose in PBS for 2 days before being snap-frozen and stored at −80 °C until sectioning. Brains were sectioned at 50 μm on a cryostat (Leica Biosystems).

Fluorescence immunohistochemistry was performed on fixed frozen 50-μm tissue sections. Briefly, sections were washed 3× PBS, underwent antigen retrieval (10 mM sodium citrate, pH6, 80 °C for 30 min), washed 3× PBS, and blocked with 5% normal donkey serum (NDS) and 0.3% TritonX-100 in PBS for 1 hour. In experiments using a biotin-labeled secondary antibody, NDS block was followed by avidin/biotin block: 0.001% avidin in PBS for 15 min, 2× PBS washes, 0.001% biotin in PBS for 15 min, 2× PBS washes. Sections were incubated in primary antibody with 2% NDS/0.3% TritonX-100 in PBS for at least 48 h at 4 °C. Sections were washed 3× PBS + 0.3% TritonX-100 and incubated in secondary antibody 1–2 h at room temperature, followed by 3× PBS washes. In all experiments using flex-GFP virus, GFP fluorescence was further amplified with an anti-GFP 488-conjugated nanobody. Experiments using biotin-labeled secondary included an additional 1 h incubation in VioBlue-streptavidin. Labeled primary antibodies were applied after secondary and incubated for at least 48 h at 4 °C. Sections were mounted on gelatinized slides, dehydrated in ascending ethanol washes, cleared in 2× xylene washes, and coverslipped for imaging. Primary antibodies used in this study: GFAP (1:1000, chicken, Rockland #200-901-D60); Aldh1l1 (1:400, rabbit, Abcam #Ab87117); Sox9 (1:1000, rabbit, Millipore Sigma #AB5535); NeuN (1:1000, chicken, Synaptic Systems #266 006); CD31 (1:100, rat, BD Biosciences #550274); V5 (1:400, human, Absolute Antibodies #AB00136-10.0); Myc (1:400, 488-labeled, Biotium #20436); FLAG (1:400, 543-labeled, Biotium #20433); HA (1:100, rat, Roche #11-867-423), and GFP (1:500, 488-labeled nanobody, ChromoTek #GBA488-100). Secondary antibodies: 1:1000, Jackson ImmunoResearch, donkey anti-rabbit (488: 711-546-152; Cy3: 711-166-152; 647: 711-606-152), donkey anti-chicken (488: 703-546-155; Cy3: 703-166-155); donkey anti-rat (Cy3: 712-166-153; 647: 712-606-153); donkey anti-human (488: 709-546-159; 647: 709-606-149); 1:250, VioBlue-streptavidin (Miltenyi Biotec, 130-106-933).

Images were analyzed using Imaris software (Oxford Instruments, v9.9) to identify and colocalize virally transduced cells. For the majority of systemic injection experiments, analysis was restricted to the cortex and included at least two sagittal sections and 2.5-mm anterior–posterior spread; for light-inducible iCreV and stroke experiments, samples were sectioned coronally to allow comparison to the non-induced/non-lesioned hemisphere. Direct intracortical virus injection tissue was sectioned coronally; analysis was restricted to the cortex and the core of the injected area and included at least two sections. GFAP percentage area covered was analyzed using Fiji Is Just ImageJ[55]. Briefly, a constant threshold was applied to maximum

**Table 1 | Plasmids deposited at Addgene**

| Promoter | Name | Transgene | Function | Addgene ID |
|---|---|---|---|---|
| GfaABC1D | AAV-GfaABC1D-Cre-4x6T | Cre | Astrocyte-selective Cre | 196410 |
| | AAV-GfaABC1D-ERCreER-4x6T | ERT2-Cre-ERT2 | Astrocyte-selective ERT2CreERT2; tamoxifen-inducible | 196411 |
| | AAV-GfaABC1D-iCreV-4x6T | iCreV | Astrocyte-selective ERT2CreERT2; light-inducible | 196412 |
| | AAV-GfaABC1D-DreO-4x6T | DreO | Astrocyte-selective Dre | 196413 |
| | AAV-GfaABC1D-lck-smFLAG-4x6T-WPRE | lck-smFLAG | Astrocyte-selective, membrane-targeted FLAG spaghetti monster | 196414 |
| | AAV-GfaABC1D-lck-smMyc-4x6T-WPRE | lck-smMyc | Astrocyte-selective, membrane-targeted Myc spaghetti monster | 196415 |
| | AAV-GfaABC1D-lck-smV5-4x6T-WPRE | lck-smV5 | Astrocyte-selective, membrane-targeted V5 spaghetti monster | 196416 |
| | AAV-GfaABC1D-MCS--4x6T-WPRE | (none) | Empty vector to add gene of interest for astrocyte-selective expression | 196417 |
| CAG | AAV-CAG-flex-GFP-4x6T | Cre-inducible GFP | Astrocyte-selective, Cre-inducible GFP | 196418 |
| | AAV-CAG-flex-lck-smV5-4x6T | Cre-inducible lck-smV5 | Astrocyte-selective, Cre-inducible, membrane-targeted V5 spaghetti monster | 196419 |
| | AAV-CAG-flex-MCS--4x6T | (none) | Empty vector to add gene of interest for Cre-inducible, astrocyte-selective expression | 196420 |
| | AAV-CAG-dDIO-lck-smMyc-4x6T | Dre-inducible lck-smMyc | Astrocyte-selective, Dre-inducible, membrane-targeted Myc spaghetti monster | 196421 |
| | AAV-CAG-dDIO-MCS--4x6T | (none) | Empty vector to add gene of interest for Dre-inducible, astrocyte-selective expression | 196422 |
| CAG, no 4x6T | AAV-CAG-lck-smV5 | Cre-inducible lck-smV5 | Cre-inducible, membrane-targeted V5 spaghetti monster; generic expression pattern | 196423 |

intensity projection images. The area of HA$^+$ RiboTag cells in cortex was outlined as a region of interest (ROI), and this outlined ROI was applied to the GFAP channel. Thresholded GFAP$^+$ area as a percentage of ROI was measured. Analysis included two contralateral sections from each of four animals per RiboTag cohort; ipsilateral tissue was used in transcriptomic analysis.

**Ribosomally loaded mRNA isolation, sequencing, and analysis**
Animals were exposed to terminal levels of isofluorane and transcardially perfused with 20 mL cold PBS. The brain was rapidly removed, placed in a pre-chilled coronal brain matrix (Zivic Instruments) on ice, and divided into left and right hemispheres. The right hemisphere was placed in 4% PFA for 24 h and processed for immunohistochemistry. The left hemisphere was further divided into 1 mm coronal sections in the brain matrix. Relevant brain slices based on viral spread patterns (A/P + 0.6 mm to −1.8 mm) were transferred to a dissection microscope in cold PBS on ice and the cortex was dissected (-M/L 1.0 mm to 3.0 mm). Tissue was transferred to 1.5 ml RNase/Dnase-free tubes, weighed, snap-frozen, and stored at −80 °C.

For immunoprecipitation, tissue was thawed on ice and homogenized in homogenization buffer at 2–3% w/v for 30 s using a motorized tissue grinder (Fisher Scientific). Homogenization buffer: 50 mM Tris pH 7.4, 100 mM KCl, 12 mM MgCl$_2$, 1% Igepal CA-630; supplemented with 1 mM DTT, 1× protease inhibitor cocktail (Sigma Aldrich), 200 U/ml RNAsin (Promega), 100 µg/ml cycloheximide, 1 mg/ml heparin[56]; all samples were processed in parallel to minimize batch effects. Samples were centrifuged for 10 min at 10,000 × $g$ at 4 °C and the supernatant was collected. In total, 50 µl per sample was reserved for input sample and stored at −80 °C until RNA extraction. The remaining sample was incubated for 4 h at 4 °C with anti-HA antibody (Biolegend #901502) at a final concentration of 10 µg/ml. Protein A/G-conjugated magnetic beads (Pierce) were rinsed with homogenization buffer, then incubated with antibody/tissue homogenate sample overnight at 4 °C, shaking. The following day, the magnetic beads were washed three times with high salt buffer. High salt buffer: 50 mM Tris pH 7.4, 300 mM KCl, 12 mM MgCl$_2$, 1% Igepal CA-630, 1 mM DTT, 100 µg/ml cycloheximide. After the final wash, beads were incubated with 350µl of RNeasy RLT lysis buffer before RNA was extracted using the RNeasy Micro Plus kit (Qiagen).

Immunoprecipitated pull-down RNA and the corresponding input RNA were examined on RNA Pico chips (Agilent) before use. All of RNA Integrity Numbers (RIN) are above 8. cDNA libraries were prepared with SMART-Seq v4 RNA Ultra Low Input (Takara) + Nextera XT (Illumina). Paired-end 100-bp sequences were generated over 1 lane by NovaSeq6000 using S4FC (Illumina).

After demultiplexing samples, we obtained between 73 and 109 million reads per sample (average: 88 M). Quality control was performed on base qualities and nucleotide composition of sequences. Alignment to the *M. musculus* (mm10) refSeq (refFlat) reference gene annotation was performed using the STAR (v2.7.5c) with default parameters. Additional QC was performed after the alignment using PicardTools. Average uniquely mapped read rate was 86.2 ± 0.9(SD)%. Total counts of read-fragments aligned to candidate gene regions were derived using HTSeq program (v0.12.4) and used as a basis for the quantification of gene expression.

Genes with CPM > 0.25 in at least three samples were selected for this analysis. Differential expression analysis was performed using the Limma-voom Bioconductor package (v3.50.1) and differentially expressed genes were selected based on FDR < 0.05 (false discovery rate, Benjamini Hochberg-adjusted $P$ values). Paired-analysis mode was used for IP vs Input.

To evaluate cell-type-specific IP-vs-input enrichment, cell-type specific genes were identified for astrocytes, neurons, endothelial cells, microglia, and oligodendrocytes[46,57]. To evaluate genes that were differentially expressed between IP samples from viral cohorts vs

Aldh1l1-CreER mice, genes were filtered for average FPKM > 1 in all IP samples and those with an FDR < 0.05 were extracted. Up- and down-regulated gene lists from each viral cohort were evaluated for representation in the top 1000 most specific genes in astrocytes, endothelial cells, microglia, neurons, and oligodendrocytes using mouse transcriptomic datasets[35]; overlap among upregulated neuronal genes across cohorts was assessed and visualized using BioVenn[58]. For Gene set Enrichment Analysis (GSEA), IP viral cohort vs Aldh1l1-CreER genes were sorted by (log2FC*-log10 pVal) to incorporate both the magnitude of the change and the statistical significance and then used as input for a preranked weighted analysis[36]. Reference gene sets were obtained for virally infected astrocytes[37–39] and stab wound astrocytes[40] and from the Molecular Signatures Database[59] for GO and were filtered for gene sets with FDR < 0.05. For visualization purposes, gene sets with an FDR = 0 were reassigned a score of 0.0001, reflecting a value lower than the lowest non-0 score. Potential miRNA target genes were identified using TarBase v.8[41]; all *Mus musculus* target genes for each miR in the 4x6T cassette (miR-124, miR126, miR137, miR-329, miR-369, miR-431) were compiled and used to filter the complete gene list of both input and IP samples. For both IP and input, genes differentially regulated between either 4x6T cohort and Aldh1l1-CreER samples but not differentially regulated between AAV2/5 and Aldh1l1-CreER samples were extracted.

## Statistics and reproducibility

Statistical analysis of repeated measures were performed in Prism 8 (GraphPad). Data were subjected to normality testing (Shapiro–Wilk test); all data except Fig. 4a passed the normality test and were analyzed using two-tailed *t* tests or one-way ANOVAs with Tukey's or Holm–Sidak's multiple comparisons tests. Figure 4a failed normality and data were analyzed using the nonparametric Kruskal–Wallis test with Dunn's multiple comparisons test. *P* values <0.05 were considered statistically significant and are reported in figure legends. All immunohistochemistry analyses were repeated in at least three mice and represent at least two sections per mouse spread along the viral transduction axis.

## Reporting summary

Further information on research design is available in the Nature Portfolio Reporting Summary linked to this article.

## Data availability

The Ribotag input and immunoprecipitated RNAseq datasets generated during the current study are available on GEO, accession GSE226366. Due to their large size, imaging datasets are available upon request; source data for imaging datasets are provided with this paper. Source data are provided with this paper.

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

## Acknowledgements

This work was supported by the Dr. Miriam and Sheldon G. Adelson Medical Research Foundation (to S.T.C. and R.K.) and the American Federation of Aging Research (to A.J.G.). We thank the members of S.T.C.'s laboratory for helpful discussions, Shutong Hou and Christine Hakobyan for assistance with tissue processing, and Qing Wang for assistance with RNA library preparation. This project involved a wide variety of plasmids and DNA constructs; we thank Gilles Bonvento for kindly sharing the miR124T cassette, and Addgene and their depositors for providing all other plasmids that made this work possible. Schematic diagrams created with Biorender.com.

## Author contributions

A.J.G. and S.T.C. designed the experiments. A.J.G. constructed the plasmids, packaged the viruses, performed the experiments, and ana-lyzed the results. R.K. performed transcriptomic data analysis. M.V.S. provided reagents, contributed to the design of experiments, and con-tributed to drafting the manuscript. A.J.G. and S.T.C. wrote the manu-script with contributions from all authors.

## Competing interests

The authors declare no competing interests.
