## [Peer Review File · Nature Communications]

A toolbox of astrocyte-specific, serotype-independent adeno-associated viral vectors using microRNA targeting sequences.REVIEWER COMMENTS

Reviewer #1 (Remarks to the Author):

It has been notoriously difficult to achieve highly selective expression of recombinases in specific types of cells in the brain using AAVs due to how little Cre expression is required for recombination. In this work Gleichman et al. present a set of useful tools for achieving highly specific, AAV-mediated Cre expression from recombinant AAV vectors administered systemically using AAV-PHP.eB or intraparenchymally using several AAV capsids. Overall the study is carefully performed and thorough. These vectors provide clear advantages over previously published strategies that rely on the GFAP or GfABC1D regulatory elements alone. The manuscript and data provides most of the critical information other scientists would need to know to make effective use of the tools. However, several major and minor points should be addressed before the work is acceptable for publication.

Major concerns:

There is no direct, within experiment comparison of the 4x6T construct and the previously published miR124T.4x construct, which also seems quite specific based on the single experiment in 4D. It's not really clear how much value these other sites add.

As part of Figure 6, the authors should include side-by-side matched experimental images of SM reporter constructs +/- the 4x6T element in systemically delivered AAV-PHP.eB to allow readers to assess the relative efficiencies and strength of expression between the control and 4x6T constructs. Does 4x6T reduce the level of expression and/or the fraction of cells with detectable expression of the transgene? This is important as the authors only provide tools with recombinases, but other researchers will likely want to use these elements for other uses where expression requirements may be different (e.g., gene editing or other reporters).

Minor concerns:

Title: the authors should remove the words serotype-independent from the title and other places in the manuscript (line 47). It would be better to state that the approach is compatible with different capsids.

The authors regularly use the term infected or infection by the recombinant AAV. Transduced and transduction would be more appropriate given that this is not an infectious virus.

Line 23 4x6T should read "increases astrocyte specificity" rather than "increases astrocytic expression"

The abstract should clarify that the 4x6T was applied in vectors that also use GfABC1D to achieve partial astrocyte specificity.

Line 37-38. The statement about AAV-PHP.eB is confusing.

Line 45. The authors should not refer to gene therapies in this case. It would be preferable to more specifically discuss these studies as astrocyte-directed reprogramming efforts.

Lines 47-52. The authors should clarify that the viral DNA genome is not the target of the miRNAs. The miRNAs reduce expression in off target cells by degrading the transgene mRNA containing the 4x6T miRNA binding site cassette.

Line 53. The authors should clarify that the 4x6T is used within the recombinant AAV genome rather than the AAV plasmid.

Line 264. The authors should more clearly state what the reference astrocyte comparator is rather than using the term “a gold standard of astrocyte mRNA expression”

Lines 288-314. Most of this paragraph discusses analyses comparing AAV transduced cell transcriptomes with prior datasets of astrocyte gene expression following a variety of insults (stab wound, Zika infection, polyIC, etc...), but it does not reference any figures to help the reader assess the comparisons. It is also confusing that Fig 5f is described as part of this paragraph, but appears to be comparing the AAV transduced cells to astrocytes from untreated Aldh111-Cre mice.

The x axis label in Fig 2a (middle) is confusing.

In the plasmid in F1e, it would be helpful to provide the nucleotide length of the 4x6T.

Reviewer #2 (Remarks to the Author):

Gleichman et al. here describe a cassette of miRNA-targeting sequences to selectively de-target neurons and endothelial cells, thereby improving astrocyte specificity of AAV vectors. The paper is of general interest as this is a welcome and much-needed improvement over currently available astrocyte-targeting techniques.

I have a few points that should be addressed:

- 1) A general comment about the statistics used here is that, given the low group sizes and likelihood that data are not normally distributed, nonparametric tests should be used throughout. I would also suggest using box and whisker plots (with individual data points included) throughout.
- 2) Fig. 1d compares one group (“no miR”) to other groups using a series of repetitive t-tests, but a single multiple comparison test (e.g. Kruskal-Wallis followed by a multiple comparisons test) should be used instead.
- 3) Fig. 1f and 4a show comparisons of groups each within the 98-99% range, and the legends claim statistical significance by one-way ANOVA. It is unclear what is compared here. I presume that differences between 99.0032%, 99.7549% and 99.8087% as in Fig. 1f (and n = 3 or 4 per group) are not significant?
- 4) The claim of specificity for astrocytes in Fig. 1 and others is mostly based on the Cre-mediated expression of fluorescent reporter genes such as GFP. However, GFP in particular is prone to post-fixation quenching etc., and it appears that anti-GFP antibodies were not used to confirm the presence or absence of GFP. Data in Fig. 1 and others using GFP should be verified using anti-GFP immunohistochemistry.

- 5) Fig. 2a and others report the specificity for astrocytes, but what about the efficacy, i.e. out of all astrocytes in a given FOV, how many were transduced/labeled? This should be reported and compared to “traditional” viruses such as PHP without miRNA.
- 6) The apparent specificity for astrocytes in the dentate gyrus, with only few labeled radial glia, is interesting. Was the high-titer or low-titer PHP injection used? If the latter, is radial glia labeling more frequent with high-titer injections? This would be important to know given the potential benefit over other approaches implicated here.
- 7) Moreover, since radial glia express most if not all known astroglial promoters, can the authors speculate on the reason for the sparse radial glia labeling? Is miRNA-induced radial glia de-targeting a possibility?
- 8) Stroke data and Fig. 3c, neural progenitor cells may also express Aldh1l1 (Foo et al., *Glia* 2013); this should be acknowledged.

Thank you for the helpful feedback on our manuscript, “A toolbox of astrocyte-specific, serotype-independent adeno-associated viral vectors using microRNA targeting sequences.” We have revised the manuscript in accordance with reviewer comments.

Reviewer 1:

Major points:

There is no direct, within experiment comparison of the 4x6T construct and the previously published miR124T.4x construct, which also seems quite specific based on the single experiment in 4D. It's not really clear how much value these other sites add.

We thank the reviewer for this point and have revised Fig 1f to include this direct comparison. We have also revised the text to make this point more clear.

As part of Figure 6, the authors should include side-by-side matched experimental images of SM reporter constructs +/- the 4x6T element in systemically delivered AAV-PHP.eB to allow readers to assess the relative efficiencies and strength of expression between the control and 4x6T constructs. Does 4x6T reduce the level of expression and/or the fraction of cells with detectable expression of the transgene?

We appreciate this valuable suggestion; Fig 6 now includes this direct matched comparison. The 4x6T cassette did not reduce efficiency (number of transduced astrocytes); we observed no decrease in strength of expression by immunohistochemistry and have reported this in the text. Unfortunately, a more quantitative evaluation of expression with western blot or RT-PCR was not possible, due to the neuronal contamination in the absence of the 4x6T cassette. While FACS could isolate astrocytes, cell sorting also strips off the majority of the astrocytic arbor; as the Ick-tagged spaghetti monster constructs are targeted to the cell membrane, this would also largely be lost in the sorting process.

Minor points:

Title: the authors should remove the words serotype-independent from the title and other places in the manuscript (line 47). It would be better to state that the approach is compatible with different capsids.

While we appreciate the reviewer's point, we feel that the fact that this cassette improves specificity regardless of serotype is a critical point of great interest to readers and thus should be included in the title of the manuscript. We have elaborated in the abstract and body of the paper, and specified that we are using the phrase “serotype-independent” to refer to the fact that this approach is compatible with different capsids.

The authors regularly use the term infected or infection by the recombinant AAV. Transduced and transduction would be more appropriate given that this is not an infectious virus.

We have changed this terminology, except in instances in which we are referring to infectious viruses.

Line 23 4x6T should read “increases astrocyte specificity” rather than “increases astrocytic expression”

We have made this change.

The abstract should clarify that the 4x6T was applied in vectors that also use GfABC1D to achieve partial astrocyte specificity.

We have made this change.

Line 37-38. The statement about AAV-PHP.eB is confusing.

We have clarified this statement.

Line 45. The authors should not refer to gene therapies in this case. It would be preferable to more specifically discuss these studies as astrocyte-directed reprogramming efforts.

We have removed this reference.

Lines 47-52. The authors should clarify that the viral DNA genome is not the target of the miRNAs. The miRNAs reduce expression in off target cells by degrading the transgene mRNA containing the 4x6T miRNA binding site cassette.

We have made this clarification.

Line 53. The authors should clarify that the 4x6T is used within the recombinant AAV genome rather than the AAV plasmid.

We have made this clarification.

Line 264. The authors should more clearly state what the reference astrocyte comparator is rather than using the term “a gold standard of astrocyte mRNA expression”

We have made this clarification.

Lines 288-314. Most of this paragraph discusses analyses comparing AAV transduced cell transcriptomes with prior datasets of astrocyte gene expression following a variety of insults (stab wound, Zika infection, polyIC, etc...), but it does not reference any figures to help the reader assess the comparisons. It is also confusing that Fig 5f is described as part of this paragraph, but appears to be comparing the AAV transduced cells to astrocytes from untreated Aldh111-Cre mice.

We apologize for the confusion; yes, in these experiments, we compared the gene changes between Aldh111-CreERT2 Ribotag samples and Ribotag samples from AAV-transduced mice. These gene sets were then assessed for overlap with the prior datasets referenced as well as all Gene Ontology gene sets, using Gene Set Enrichment Analysis. Had any of the referenced prior datasets passed significance threshold, they would have been listed on the X axis of Fig 5f along with the GO gene sets that did show significant enrichment. We have re-phrased parts of this paragraph and included these negative data as a supplementary figure to address this confusion.

The x axis label in Fig 2a (middle) is confusing.

We have adjusted the label and the corresponding legend to minimize confusion.

In the plasmid in F1e, it would be helpful to provide the nucleotide length of the 4x6T.

We have added this information to both Fig 1e and the accompanying text.

Reviewer 2:

1) A general comment about the statistics used here is that, given the low group sizes and likelihood that data are not normally distributed, nonparametric tests should be used throughout. I would also suggest using box and whisker plots (with individual data points included) throughout.

Thank you for the comment. All data were subject to normality testing; with the exception of Figure 4a, all passed normality tests (Shapiro-Wilk). Figure 4a, therefore, has been adjusted to reflect the nonparametric Kruskal-Wallis test. This information has been added to the methodology and figure 4a legend. While we appreciate the suggestion to use box and whisker plots, given that the n's throughout the study are <10, we feel that bar graphs with individual values plotted and exact means reported are more visually informative; this is consistent with Nature Communications' guidelines.

2) Fig. 1d compares one group ("no miR") to other groups using a series of repetitive t-tests, but a single multiple comparison test (e.g. Kruskal-Wallis followed by a multiple comparisons test) should be used instead.

Thank you for the comment. We have conducted a one-way ANOVA to assess these data, given that they passed normality testing (Shapiro-Wilk, see above response); the figure and legend now reflect this.

3) Fig. 1f and 4a show comparisons of groups each within the 98-99% range, and the legends claim statistical significance by one-way ANOVA. It is unclear what is compared here. I presume that differences between 99.0032%, 99.7549% and 99.8087% as in Fig. 1f (and $n = 3$ or 4 per group) are not significant?

While we agree that it is unusual to find statistical significance in this range, the data are quite consistent within groups and the statistical testing reflects this. This was conducted with a one-way ANOVA with Tukey's multiple comparisons, as reported in the figure legend. For example, the standard deviations in Fig 1f are: 0.1123 (flexGFP with anti-GFP antibody), 0.08642 (Ai14), 0.07586 (flexV5-4x6T). To make this more visually clear, we have adjusted the Y-axes of Fig 1f and 4a.

4) The claim of specificity for astrocytes in Fig. 1 and others is mostly based on the Cre mediated expression of fluorescent reporter genes such as GFP. However, GFP in particular is prone to post-fixation quenching etc., and it appears that anti-GFP antibodies were not used to confirm the presence or absence of GFP. Data in Fig. 1 and others using GFP should be verified using anti-GFP immunohistochemistry.

We have replicated the data from the manuscript that relied upon GFP fluorescence with anti-GFP immunostaining (Fig 1a, d, f); the results are highly consistent with the results obtained with GFP fluorescence.

5) Fig. 2a and others report the specificity for astrocytes, but what about the efficacy, i.e. out of all astrocytes in a given FOV, how many were transduced/labeled? This should be reported and compared to "traditional" viruses such as PHP without miRNA.

Thank you for the comment. Fig 2a includes the efficiency, labeled in the original manuscript as "% of Sox9+ astrocytes labeled." We have re-labeled this figure and the accompanying legend to be more clear. We have further included a direct comparison of specificity and efficiency for constructs with and without the 4x6T cassette in Figure 6a.

6) The apparent specificity for astrocytes in the dentate gyrus, with only few labeled radial glia, is interesting. Was the high-titer or low-titer PHP injection used? If the latter, is radial glia labeling more frequent with high-titer injections? This would be important to know given the potential benefit over other approaches implicated here.

The data originally shown in Fig 3b reflect mid-titer injections; to more fully understand hippocampal injection patterns, we returned to high-titer injections in Ai14 adult mice, which continue to show very low levels of radial glial labeling. We have included these data in the current manuscript (Fig 3c).

7) Moreover, since radial glia express most if not all known astroglial promoters, can the authors speculate on the reason for the sparse radial glia labeling? Is miRNA-induced radial glia de-targeting a possibility?

The reviewer raises an interesting point; as miR124 in particular is known to be highly expressed in progenitor cells, it is possible that the 4x6T cassette promotes transgene degradation in progenitors, but is not effective enough to fully de-target radial glia, particularly when injected at early postnatal stages with high numbers of radial glia present. We have added this speculation to the discussion.

8) Stroke data and Fig. 3c, neural progenitor cells may also express Aldh111 (Foo et al., *Glia* 2013); this should be acknowledged.

While the work of Foo et al found Aldh111 can be present in NPCs primarily using BAC Aldh111 mice, more recent single-cell data (Hochgerner et al, *Nat Neurosci* 2018) suggest that Aldh111 expression is lower in radial glia vs astrocytes and largely absent in neuroblasts, which are the neural progenitor population largely found in the peri-infarct after stroke. This has been functionally recapitulated by data that show that Aldh111-CreER transgenic mice show very little recombination in neural stem cells in the dentate gyrus when recombination is induced starting at postnatal day 56 (Beyer et al., *Front. Neurosci* 2021). We have elaborated on this point in the manuscript to minimize confusion.

REVIEWERS' COMMENTS

Reviewer #1 (Remarks to the Author):

The authors have address my prior concerns. I have only new minor comment:

Several of the figures/figure legends are missing important experimental information. The authors should make sure they have included the age and route of administration, the dose(s), and the length of time between dosing and tissue collection for each figure/experiment.

Reviewer #2 (Remarks to the Author):

All my points have been sufficiently addressed and I have no further comments.

We have revised the manuscript in accordance with the additional reviewer comments.

Reviewer 1:

The authors should make sure they have included the age and route of administration, the dose(s), and the length of time between dosing and tissue collection for each figure/experiment.

We have added this information to the figure legends. These changes are highlighted in yellow in the legends.